# PreDiff: Precipitation Nowcasting with Latent Diffusion Models

**Zhihan Gao**[*]
Hong Kong University of Science and Technology
zhihan.gao@connect.ust.hk

**Xingjian Shi**[†]
Boson AI
xshiab@connect.ust.hk

**Boran Han**
AWS
boranhan@amazon.com

**Hao Wang**
AWS AI Labs
howngz@amazon.com

**Xiaoyong Jin**
Amazon
jxiaoyon@amazon.com

**Danielle Maddix**
AWS AI Labs
dmmaddix@amazon.com

**Yi Zhu**[†]
Boson AI
yi@boson.ai

**Mu Li**[†]
Boson AI
mu@boson.ai

**Yuyang Wang**
AWS AI Labs
yuyawang@amazon.com

## Abstract

Earth system forecasting has traditionally relied on complex physical models that are computationally expensive and require significant domain expertise. In the past decade, the unprecedented increase in spatiotemporal Earth observation data has enabled data-driven forecasting models using deep learning techniques. These models have shown promise for diverse Earth system forecasting tasks. However, they either struggle with handling uncertainty or neglect domain-specific prior knowledge; as a result, they tend to suffer from averaging possible futures to blurred forecasts or generating physically implausible predictions. To address these limitations, we propose a two-stage pipeline for probabilistic spatiotemporal forecasting: 1) We develop PreDiff, a conditional latent diffusion model capable of probabilistic forecasts. 2) We incorporate an explicit knowledge alignment mechanism to align forecasts with domain-specific physical constraints. This is achieved by estimating the deviation from imposed constraints at each denoising step and adjusting the transition distribution accordingly. We conduct empirical studies on two datasets: $N$-body MNIST, a synthetic dataset with chaotic behavior, and SEVIR, a real-world precipitation nowcasting dataset. Specifically, we impose the law of conservation of energy in $N$-body MNIST and anticipated precipitation intensity in SEVIR. Experiments demonstrate the effectiveness of PreDiff in handling uncertainty, incorporating domain-specific prior knowledge, and generating forecasts that exhibit high operational utility.

## 1 Introduction

Earth's intricate climate system significantly influences daily life. Precipitation nowcasting, tasked with delivering accurate rainfall forecasts for the near future (e.g., 0-6 hours), is vital for decision-making across numerous industries and services. Recent advancements in data-driven deep learning (DL) techniques have demonstrated promising potential in this field, rivaling conventional numerical methods [8, 5] with their advantages of being more skillful [5], efficient [37], and scalable [3]. However, accurately predicting the future rainfall remains challenging for data-driven algorithms.

---

[*]Work conducted during an internship at Amazon. [†]Work conducted while at Amazon.

NeurIPS 2023 AI for Science Workshop.

The state-of-the-art Earth system forecasting algorithms [47, 61, 41, 37, 8, 69, 2, 29, 3] typically generate blurry predictions. This is caused by the high variability and complexity inherent to Earth's climatic system. Even minor differences in initial conditions can lead to vastly divergent outcomes that are difficult to predict. Most methods adopt a point estimation of the future rainfall and are trained by minimizing pixel-wise loss functions (e.g., mean-squared error). These methods lack the capability of capturing multiple plausible futures and will generate blurry forecasts which lose important operational details. Therefore, what are needed instead are probabilistic models that can represent the uncertainty inherent in stochastic systems. The probabilistic models can capture multiple plausible futures, generating diverse high-quality predictions that better align with real-world data.

The emergence of diffusion models (DMs) [22] has enabled powerful probabilistic frameworks for generative modeling. DMs have shown remarkable capabilities in generating high-quality images [40, 45, 43] and videos [15, 23]. As a likelihood-based model, DMs do not exhibit mode collapse or training instabilities like GANs [10]. Compared to autoregressive (AR) models [53, 46, 63, 39, 65] that generate images pixel-by-pixel, DMs can produce higher resolution images faster and with higher quality. They are also better at handling uncertainty [62, 34, 57–59] without drawbacks like exposure bias [13] in AR models. Latent diffusion models (LDMs) [42, 52] further improve on DMs by separating the model into two phases, only applying the costly diffusion in a compressed latent space. This alleviates the computational costs of DMs without significantly impairing performance.

Despite DMs' success in image and video generation [42, 15, 66, 36, 32, 56], its application to precipitation nowcasting and Earth system forecasting is in early stages [16]. One of the major concerns is that this purely data-centric approach lacks constraints and controls from prior knowledge about the dynamic system. Some spatiotemporal forecasting approaches have incorporated domain knowledge by modifying the model architecture or adding extra training losses [11, 1, 37]. This enables them to be aware of prior knowledge and generate physically plausible forecasts. However, these approaches still face challenges, such as requiring to design new model architectures or retrain the entire model from scratch when constraints change. More detailed discussions on related works are provided in Appendix B.

Inspired by recent success in controllable generative models [68, 24, 4, 33, 6], we propose a general two-stage pipeline for training data-driven Earth system forecasting model. 1) In the first stage, we focus on capturing the intrinsic semantics in the data by training an LDM. To capture Earth's long-term and complex changes, we instantiate the LDM's core neural network as a UNet-style architecture based on Earthformer [8]. 2) In the second stage, we inject prior knowledge of the Earth system by training a knowledge alignment network that guides the sampling process of the LDM. Specifically the alignment network parameterizes an energy function that adjusts the transition probabilities during each denoising step. This encourages the generation of physically plausible intermediate latent states while suppressing those likely to violate the given domain knowledge. We summarize our main contributions as follows:

- We introduce a novel LDM based model *PreDiff* for precipitation nowcasting.

- We propose a general two-stage pipeline for training data-driven Earth system forecasting models. Specifically, we develop *knowledge alignment* mechanism to guide the sampling process of PreDiff. This mechanism ensures that the generated predictions align with domain-specific prior knowledge better, thereby enhancing the reliability of the forecasts, without requiring any modifications to the trained PreDiff model.

- Our method achieves state-of-the-art performance on the $N$-body MNIST [8] dataset and attains state-of-the-art perceptual quality on the SEVIR [55] dataset.

## 2  Method

We follow [47, 48, 55, 1, 8] to formulate precipitation nowcasting as a spatiotemporal forecasting problem. The $L_{\text{in}}$-step observation is represented as a spatiotemporal sequence $y = [y^j]_{j=1}^{L_{\text{in}}} \in \mathbb{R}^{L_{\text{in}} \times H \times W \times C}$, where $H$ and $W$ denote the spatial resolution, and $C$ denotes the number of measurements at each space-time coordinate. Probabilistic forecasting aims to model the conditional probabilistic distribution $p(x|y)$ of the $L_{\text{out}}$-step-ahead future $x = [x^j]_{j=1}^{L_{\text{out}}} \in \mathbb{R}^{L_{\text{out}} \times H \times W \times C}$, given the observation $y$. In what follows, we will present the parameterization of $p(x|y)$ by a controllable LDM.

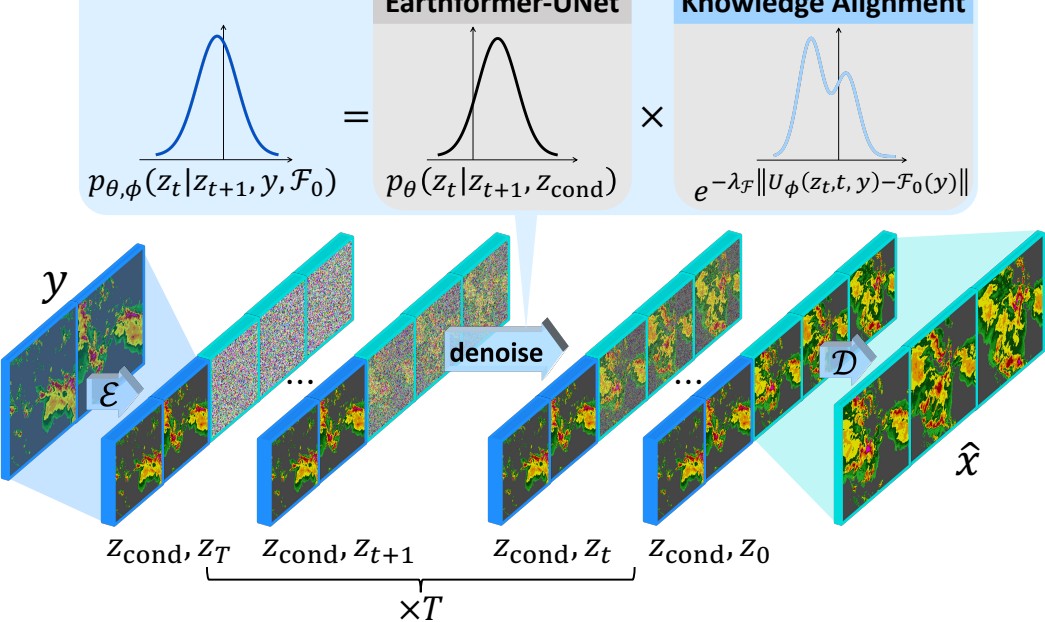

Figure 1: **Overview of PreDiff inference with knowledge alignment.** An observation sequence $y$ is encoded into a latent context $z_{\text{cond}}$ by the frame-wise encoder $\mathcal{E}$. The latent diffusion model $p_\theta(z_t|z_{t+1}, z_{\text{cond}})$, which is parameterized by an Earthformer-UNet, then generates the latent future $z_0$ by autoregressively denoising Gaussian noise $z_T$ conditioned on $z_{\text{cond}}$. It takes the concatenation of the latent context $z_{\text{cond}}$ (in the blue border) and the previous-step noisy latent future $z_{t+1}$ (in the cyan border) as input, and outputs $z_t$. The transition distribution of each step from $z_{t+1}$ to $z_t$ can be further refined as $p_{\theta,\phi}(z_t|z_{t+1}, y, \mathcal{F}_0)$ via knowledge alignment, according to auxiliary prior knowledge. This denoising process iterates from $t = T$ to $t = 0$, resulting in a denoised latent future $z_0$. Finally, $z_0$ is decoded back to pixel space by the frame-wise decoder $\mathcal{D}$ to produce the final prediction $\widehat{x}$. (Best viewed in color).

## 2.1 Preliminary: Diffusion Models

Diffusion models (DMs) learn the data distribution $p(x)$ by training a model to reverse a predefined noising process that progressively corrupts the data. Specifically, the noising process is defined as $q(x_t|x_{t-1}) = \mathcal{N}(x_t; \sqrt{\alpha_t}x_{t-1}, (1 - \alpha_t)I), 1 \leq t \leq T$, where $x_0 \sim p(x)$ is the true data, and $x_T \sim \mathcal{N}(0, I)$ is random noise. The coefficients $\alpha_t$ follow a fixed schedule over the timesteps $t$. DMs factorize and parameterize the joint distribution over the data $x_0$ and noisy latents $x_i$ as $p_\theta(x_{0:T}) = p(x_T) \prod_{t=1}^{T} p_\theta(x_{t-1}|x_t)$, where each step of the reverse denoising process is a Gaussian distribution $p_\theta(x_{t-1}|x_t) = \mathcal{N}(\mu_\theta(x_t, t), \Sigma_\theta(x_t, t))$, which is trained to recover $x_{t-1}$ from $x_t$.

To apply DMs for spatiotemporal forecasting, $p(x|y)$ is factorized and parameterized as $p_\theta(x|y) = \int p_\theta(x_{0:T}|y)dx_{1:T} = \int p(x_T) \prod_{t=1}^{T} p_\theta(x_{t-1}|x_t, y)dx_{1:T}$, where $p_\theta(x_{t-1}|x_t, y)$ represents the conditional denoising transition with the condition $y$.

## 2.2 Conditional Diffusion in Latent Space

To improve the computational efficiency of DM training and inference, our *PreDiff* follows LDM to adopt a two-phase training that leverages the benefits of lower-dimensional latent representations. The two sequential phases of the PreDiff training are: 1) Training a frame-wise variational autoencoder (VAE) [28] that encodes pixel space into a lower-dimensional latent space, and 2) Training a conditional DM that generates predictions in this acquired latent space.

**Frame-wise autoencoder.** We follow [7] to train a frame autoencoder using a combination of the pixel-wise loss (e.g. L2 loss) and an adversarial loss. Different from [7], we exclude the perceptual loss since there are no standard pretrained models for perception on Earth observation data. Specifically, the encoder $\mathcal{E}$ is trained to encode a data frame $x^j \in \mathbb{R}^{H \times W \times C}$ to a latent representation $z^j = \mathcal{E}(x^j) \in \mathbb{R}^{H_z \times W_z \times C_z}$. The decoder $\mathcal{D}$ learns to reconstruct the data frame $\widehat{x}^j = \mathcal{D}(z^j)$ from

the encoded latent. We denote $z \sim p_{\mathcal{E}}(z|x) \in \mathbb{R}^{L \times H_z \times W_z \times C_z}$ as equivalent to $z = [z^j] = [\mathcal{E}(x^j)]$, representing encoding a sequence of frames in pixel space into a latent spatiotemporal sequence. And $x \sim p_{\mathcal{D}}(x|z)$ denotes decoding a latent spatiotemporal sequence.

**Latent diffusion.** With the context $y$ being encoded by the frame-wise encoder $\mathcal{E}$ into the learned latent space as $z_{\text{cond}} \in \mathbb{R}^{L_{\text{in}} \times H_z \times W_z \times C_z}$ as (1). The conditional distribution $p_\theta(z_{0:T}|z_{\text{cond}})$ of the latent future $z_i \in \mathbb{R}^{L_{\text{out}} \times H_z \times W_z \times C_z}$ given $z_{\text{cond}}$ is factorized and parameterized as (2):

$$z_{\text{cond}} \sim p_{\mathcal{E}}(z_{\text{cond}}|y), \tag{1}$$

$$p_\theta(z_{0:T}|z_{\text{cond}}) = p(z_T) \prod_{t=1}^{T} p_\theta(z_{t-1}|z_t, z_{\text{cond}}). \tag{2}$$

where $z_T \sim p(z_T) = \mathcal{N}(0, I)$. As proposed by [22, 45], an equivalent parameterization is to have the DMs learn to match the transition noise $\epsilon_\theta(z_t, t)$ of step $t$ instead of directly predicting $z_{t-1}$. The training objective of PreDiff is simplified as shown in (3):

$$L_{\text{CLDM}} = \mathbb{E}_{(x,y),t,\epsilon \sim \mathcal{N}(0,I)} \|\epsilon - \epsilon_\theta(z_t, t, z_{\text{cond}})\|_2^2. \tag{3}$$

where $(x, y)$ is a sampled context sequence and target sequence data pair, and given that, $z_t \sim q(z_t|z_0)p_{\mathcal{E}}(z_0|x)$ and $z_{\text{cond}} \sim p_{\mathcal{E}}(z_{\text{cond}}|y)$.

**Instantiating $p_\theta(z_{t-1}|z_t, z_{\text{cond}})$.** Compared to images, modeling spatiotemporal observation data in precipitation nowcasting poses greater challenges due to their higher dimensionality. We propose replacing the UNet backbone in LDM [42] with *Earthformer-UNet*, derived from Earthformer's encoder [8], which is known for its ability to model intricate and extensive spatiotemporal dependencies in the Earth system.

Earthformer-UNet adopts a hierarchical UNet architecture with self cuboid attention [8] as the building blocks, excluding the bridging cross-attention in the encoder-decoder architecture of Earthformer. More details of the architecture design of Earthformer-UNet are provide in Appendix C.1. We find Earthformer-UNet to be more stable and effective at modeling the transition distribution $p_\theta(z_{t-1}|z_t, z_{\text{cond}})$. It takes the concatenation of the encoded latent context $z_{\text{cond}}$ and the noisy latent future $z_t$ along the temporal dimension as input, and predicts the one-step-ahead noisy latent future $z_{t-1}$ (in practice, the transition noise $\epsilon$ from $z_t$ to $z_{t-1}$ is predicted as shown in (3)).

## 2.3 Incorporating Knowledge Alignment

Though DMs hold great promise for diverse and realistic generation, the generated predictions may violate physical constraints, or disregard domain-specific prior knowledge, thereby fail to give plausible and non-trivial results [14, 44]. One possible reason for this is that DMs are not necessarily trained on data full compliant with domain knowledge. When trained on such data, there is no guarantee that the generations sampled from the learned distribution will re-

---

**Algorithm 1** One training step of the knowledge alignment network $U_\phi$

---
1: $(x, y)$ sampled from data
2: $t \sim \text{Uniform}(0, T)$
3: $z_t \sim q(z_t|z_0)p_{\mathcal{E}}(z_0|x)$
4: $L_U \leftarrow \|U_\phi(z_t, t, y) - \mathcal{F}(x, y)\|$

---

main physically realizable. The causes may also stem from the stochastic nature of chaotic systems, the approximation error in denoising steps, etc.

To address this issue, we propose *knowledge alignment* to incorporate auxiliary prior knowledge:

$$\mathcal{F}(\widehat{x}, y) = \mathcal{F}_0(y) \in \mathbb{R}^d, \tag{4}$$

into the diffusion generation process. The knowledge alignment imposes a constraint $\mathcal{F}$ on the forecast $\widehat{x}$, optionally with the observation $y$, based on domain expertise. E.g., for an isolated physical system, the knowledge $E(\widehat{x}, \cdot) = E_0(y^{L_{\text{in}}}) \in \mathbb{R}$ imposes the conservation of energy by enforcing the generation $\widehat{x}$ to keep the total energy $E(\widehat{x}, \cdot)$ the same as the last observation $E_0(y^{L_{\text{in}}})$. The violation $\|\mathcal{F}(\widehat{x}, y) - \mathcal{F}_0(y)\|$ quantifies the deviation of a prediction $\widehat{x}$ from prior knowledge. The larger violation indicates $\widehat{x}$ diverges further from the constraints. Knowledge alignment hence aims to suppress the probability of generating predictions with large violation. Notice that even the target

futures $x$ from training data may violate the knowledge, i.e. $\mathcal{F}(x,y) \neq \mathcal{F}_0(y)$, due to noise in data collection or simulation.

Inspired by classifier guidance [4], we achieve knowledge alignment by training a knowledge alignment network $U_\phi(z_t, t, y)$ to estimate $\mathcal{F}(\hat{x}, y)$ from the intermediate latent $z_t$ at noising step $t$. The key idea is to adjust the transition probability distribution $p_\theta(z_{t-1}|z_t, z_{\text{cond}})$ in (2) during each latent denoising step to reduce the likelihood of sampling $z_t$ values expected to violate the constraints:

$$p_{\theta,\phi}(z_t|z_{t+1}, y, \mathcal{F}_0) \propto p_\theta(z_t|z_{t+1}, z_{\text{cond}}) \cdot e^{-\lambda_\mathcal{F} \|U_\phi(z_t, t, y) - \mathcal{F}_0(y)\|}, \qquad (5)$$

where $\lambda_\mathcal{F}$ is a guidance scale factor. The knowledge alignment network is trained by optimizing the objective $L_U$ in Alg. 1. According to [4], (5) can be approximated by shifting the predicted mean of the denoising transition $\mu_\theta(z_{t+1}, t, z_{\text{cond}})$ by $-\lambda_\mathcal{F} \Sigma_\theta \nabla_{z_t} \|U_\phi(z_t, t, y) - \mathcal{F}_0(y)\|$, where $\Sigma_\theta$ is the variance of the original transition distribution $p_\theta(z_t|z_{t+1}, z_{\text{cond}}) = \mathcal{N}(\mu_\theta(z_{t+1}, t, z_{\text{cond}}), \Sigma_\theta(z_{t+1}, t, z_{\text{cond}}))$. Detailed derivation is provided in Appendix D.

The training procedure of knowledge alignment is outlined in Alg. 1. The noisy latent $z_t$ for training the knowledge alignment network $U_\phi$ is sampled by encoding the target $x$ using the frame-wise encoder $\mathcal{E}$ and the forward noising process $q(z_t|z_0)$, eliminating the need for an inference sampling process. This makes the training of the knowledge alignment network $U_\phi$ independent of the LDM training. At inference time, the knowledge alignment mechanism is applied as a plug-in, without impacting the trained VAE and the LDM. This modular approach allows training lightweight knowledge alignment networks $U_\phi$ to flexibly explore various constraints and domain knowledge, without the need for retraining the entire model. This stands as a key advantage over incorporating constraints into model architectures or training losses.

## 3 Experiments

We conduct empirical studies and compare PreDiff with other state-of-the-art spatiotemporal forecasting models on a synthetic dataset $N$-body MNIST [8] and a real-world precipitation nowcasting benchmark SEVIR[2] [55] to verify the effectiveness of PreDiff in handling the dynamics and uncertainty in complex spatiotemporal systems and generating high quality, accurate forecasts. We impose data-specific knowledge alignment: **energy conservation** on $N$-body MNIST and **anticipated precipitation intensity** on SEVIR. Experiments demonstrate that PreDiff under the guidance of knowledge alignment (PreDiff-KA) is able to generate predictions that comply with domain expertise much better, without severely sacrificing fidelity. In what follows, we will present the empirical studies on SEVIR. The results on $N$-body MNIST and the corresponding analysis are provided in Appendix A.

### 3.1 SEVIR Precipitation Nowcasting

**Dataset.** The Storm EVent ImageRy (SEVIR) [55] is a spatiotemporal Earth observation dataset which consists of $384 \text{ km} \times 384 \text{ km}$ image sequences spanning over 4 hours. Images in SEVIR are sampled and aligned across five different data types: three channels (C02, C09, C13) from the GOES-16 advanced baseline imager, NEXRAD Vertically Integrated Liquid (VIL) mosaics, and GOES-16 Geostationary Lightning Mapper (GLM) flashes. The SEVIR benchmark supports scientific research on multiple meteorological applications including precipitation nowcasting, synthetic radar generation, front detection, etc. Due to computational resource limitations, we adopt a downsampled version of SEVIR for benchmarking precipitation nowcasting. The task is to predict the future VIL up to 60 minutes (6 frames) given 70 minutes of context VIL (7 frames) at a spatial resolution of $128 \times 128$, i.e. $x \in \mathbb{R}^{6 \times 128 \times 128 \times 1}$, $y \in \mathbb{R}^{7 \times 128 \times 128 \times 1}$.

**Evaluation.** Following [55, 8], we adopt the `Critical Success Index` (CSI) for evaluation, which is commonly used in precipitation nowcasting and is defined as `CSI` $= \frac{\#\text{Hits}}{\#\text{Hits}+\#\text{Misses}+\#\text{F.Alarms}}$. To count the $\#$`Hits` (truth=1, pred=1), $\#$`Misses` (truth=1, pred=0) and $\#$`F.Alarms` (truth=0, pred=1), the prediction and the ground-truth are rescaled to the range $0-255$

---

[2]Dataset is available at `https://sevir.mit.edu/`

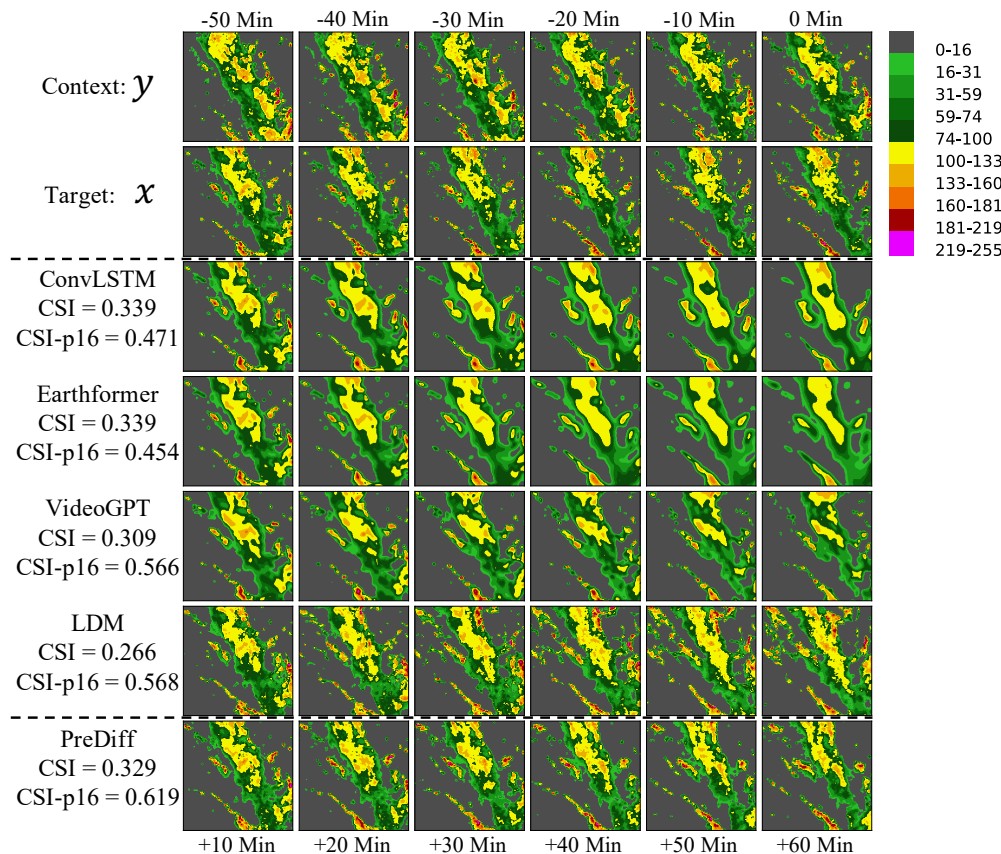

Figure 2: A set of example forecasts from baselines and PreDiff on the SEVIR test set. From top to bottom: context sequence $y$, target sequence $x$, forecasts from ConvLSTM [47], Earthformer [8], VideoGPT[65], LDM [42], PreDiff.

and binarized at thresholds $[16, 74, 133, 160, 181, 219]$. We also follow [41] to report the CSI at pooling scale $4 \times 4$ and $16 \times 16$, which evaluate the performance on neighborhood aggregations at multiple spatial scales. These pooled CSI metrics assess the models' ability to capture local pattern distributions. Additionally, we incorporate FVD [51] and continuous ranked probability score (CRPS) [9] for assessing the visual quality and uncertainty modeling capabilities of the investigated methods. Similar to Fréchet Inception Distance (FID) [20] for evaluating image generation, FVD estimates the distance between the learned distribution and the true data distribution by comparing the statistics of feature vectors extracted from the generations and the real data. The inception network used in FVD for feature extraction is pre-trained on video classification and is not specifically adapted for processing "unnatural videos" such as spatiotemporal observation data in Earth systems. Consequently, the FVD scores on SEVIR cannot be directly compared with those on natural video datasets. Nevertheless, the relative ranking of the FVD scores remains a meaningful indicator of model ability to achieve high visual quality, as FVD has shown consistency with expert evaluations across various domains beyond natural images [38, 26]. CRPS measures the discrepancy between the predicted distribution and the true distribution. When the predicted distribution collapses into a single value, as in deterministic models, CRPS reduces to Mean Absolute Error (MAE). A lower CRPS value indicates higher forecast accuracy. Scores for all involved metrics are calculated using an ensemble of eight samples from each model.

### 3.1.1 Comparison to the State of the Art

We adjust the configurations of involved baselines accordingly and tune some of the hyperparameters for adaptation on the SEVIR dataset. More implementation details of baselines are provided in Appendix C.2. The experiment results listed in Table 1 show that probabilistic spatiotemporal forecasting methods are not good at achieving high CSI scores. However, they are more powerful at capturing the patterns and the true distribution of the data, hence achieving much better FVD

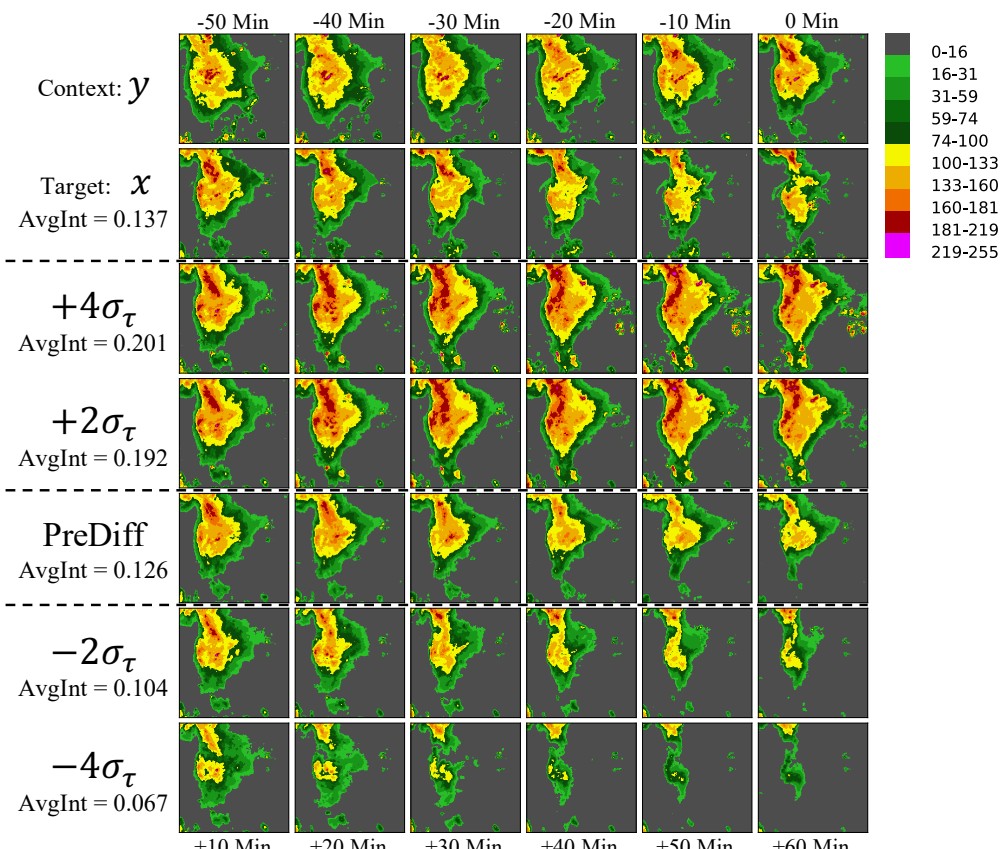

Figure 3: A set of example forecasts from PreDiff-KA, i.e., PreDiff under the guidance of anticipated average intensity. From top to bottom: context sequence $y$, target sequence $x$, forecasts from PreDiff and PreDiff-KA showcasing different levels of anticipated future intensity ($\mu_\tau + n\sigma_\tau$), where $n$ takes the values of $4, 2, -2, -4$.

scores and CSI-pool16. Qualitative results shown in Fig. 2 demonstrate that CSI is not aligned with human perceptual judgement. For such a complex system, deterministic methods give up capturing the real patterns and resort to averaging the possible futures, i.e. blurry predictions, to keep the scores from appearing too inaccurate. Probabilistic approaches, of which PreDiff is the best, though are not favored by per-pixel metrics, perform better at capturing the data distribution within a local area, resulting in higher CSI-pool16, lower CRPS, and succeed in keeping the correct local patterns, which can be crucial for recognizing weather events. More detailed quantitative results on SEVIR are provided in Appendix E.

### 3.1.2 Knowledge Alignment: Anticipated Average Intensity

Earth system observation data, such as the Vertically Integrated Liquid (VIL) data in SEVIR, are usually not physically complete, posing challenges for directly incorporating physical laws for guidance. However, with highly flexible knowledge alignment mechanism, we can still utilize auxiliary prior knowledge to guide the forecasting effectively. Specifically for precipitation nowcasting on SEVIR, we use anticipated precipitation intensity to align the generations to simulate possible extreme weather events. We denote the average intensity of a data sequence as $I(x) \in \mathbb{R}^+$. In order to estimate the conditional quantiles of future intensity, we train a simple probabilistic time series forecasting model with a parametric (Gaussian) distribution $p_\tau(I(x)|[I(y^j)]) = \mathcal{N}(\mu_\tau([I(y^j)]), \sigma_\tau([I(y^j)]))$ that predict the distribution of the average future intensity $I(x)$ given the average intensity of each context frame $[I(y^j)]_{j=1}^{L_{in}}$ (abbreviated as $[I(y^j)]$). By incorporating $\mathcal{F}(\widehat{x}, y) \equiv I(\widehat{x})$ and $\mathcal{F}_0(y) \equiv \mu_\tau + n\sigma_\tau$ for knowledge alignment, PreDiff-KA gains the capability of generating forecasts for potential extreme cases, e.g., where $I(\widehat{x})$ falls outside the typical range of $\mu_\tau \pm \sigma_\tau$.

Table 1: Performance comparison on SEVIR. The `Critical Success Index`, also known as the intersection over union (IoU), is calculated at different precipitation thresholds and denoted as `CSI`-$thresh$. `CSI` reports the mean of `CSI`-$[16, 74, 133, 160, 181, 219]$. `CSI-pool`$s$ with $s = 4$ and $s = 16$ report the `CSI` at pooling scales of $4 \times 4$ and $16 \times 16$. Besides, we include the continuous ranked probability score (CRPS) for probabilistic forecast assessment, and the scores of Fréchet Video Distance (FVD) for evaluating visual quality.

| Model | #Param. (M) | Metrics | | | | |
|---|---|---|---|---|---|---|
| | | FVD ↓ | CRPS ↓ | CSI ↑ | CSI-pool4 ↑ | CSI-pool16 ↑ |
| Persistence | - | 525.2 | 0.0526 | 0.2613 | 0.3702 | 0.4690 |
| UNet [55] | 16.6 | 753.6 | 0.0353 | 0.3593 | 0.4098 | 0.4805 |
| ConvLSTM [47] | 14.0 | 659.7 | 0.0332 | 0.4185 | 0.4452 | 0.5135 |
| PredRNN [61] | 46.6 | 663.5 | 0.0306 | 0.4080 | 0.4497 | 0.5005 |
| PhyDNet [11] | 13.7 | 723.2 | 0.0319 | 0.3940 | 0.4379 | 0.4854 |
| E3D-LSTM [60] | 35.6 | 600.1 | 0.0297 | 0.4038 | 0.4492 | 0.4961 |
| Rainformer [1] | 184.0 | 760.5 | 0.0357 | 0.3661 | 0.4232 | 0.4738 |
| Earthformer [8] | 15.1 | 690.7 | 0.0304 | **0.4419** | 0.4567 | 0.5005 |
| DGMR [41] | 71.5 | 485.2 | 0.0435 | 0.2675 | 0.3431 | 0.4832 |
| VideoGPT [65] | 99.6 | 261.6 | 0.0381 | 0.3653 | 0.4349 | 0.5798 |
| LDM [42] | 438.6 | 133.0 | 0.0280 | 0.3580 | 0.4022 | 0.5522 |
| PreDiff | 220.5 | **33.05** | **0.0246** | 0.4100 | **0.4624** | **0.6244** |
| PreDiff-KA ($\in [-2\sigma_\tau, 2\sigma_\tau]$) | 229.4 | 34.18 | - | - | - | - |

Fig. 3 shows a set of generations from PreDiff and PreDiff-KA with anticipated future intensity $\mu_\tau + n\sigma_\tau$, $n \in \{-4, -2, 2, 4\}$. This qualitative example demonstrates that PreDiff is not only capable of capturing the distribution of the future, but also flexible at highlighting possible extreme cases like rainstorms and droughts with the knowledge alignment mechanism, which is crucial for decision-making and precaution.

According to Table 1, the FVD score of PreDiff-KA (34.18) is only slightly worse than the FVD score of PreDiff (33.05). This indicates that knowledge alignment effectively aligns the generations with prior knowledge while maintaining fidelity and adherence to the true data distribution.

## 4  Conclusions and Broader Impacts

In this paper, we propose PreDiff, a novel latent diffusion model for precipitation nowcasting. We also introduce a general two-stage pipeline for training DL models for Earth system forecasting. Specifically, we develop knowledge alignment mechanism that is capable of guiding PreDiff to generate forecasts in compliance with domain-specific prior knowledge. Experiments demonstrate that our method achieves state-of-the-art performance on $N$-body MNIST and SEVIR datasets.

Our work has certain limitations: 1) Benchmark datasets and evaluation metrics for precipitation nowcasting and Earth system forecasting are still maturing compared to the computer vision domain. While we utilize conventional precipitation forecasting metrics and visual quality evaluation, aligning these assessments with expert judgement remains an open challenge. 2) Effective integration of physical principles and domain knowledge into DL models for precipitation nowcasting remains an active research area. Close collaboration between DL researchers and domain experts in meteorology and climatology will be key to developing hybrid models that effectively leverage both data-driven learning and scientific theory. 3) While Earth system observation data have grown substantially in recent years, high-quality data remain scarce in many domains. This scarcity can limit PreDiff's ability to accurately capture the true distribution, occasionally resulting in unrealistic forecast hallucinations under the guidance of prior knowledge as it attempts to circumvent the knowledge alignment mechanism. Further research on enhancing the sample efficiency of PreDiff and the knowledge alignment mechanism is needed.

In conclusion, PreDiff represents a promising advance in knowledge-aligned DL for Earth system forecasting, but work remains to improve benchmarking, incorporate scientific knowledge, and boost model robustness through collaborative research between AI and domain experts.

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

# A Experiments on $N$-body MNIST: Digits Motion Forecasting

**Dataset.** The Earth is a chaotic system with complex dynamics. The real-world Earth observation data, such as radar echo maps and satellite imagery, are usually not physically complete. We are unable to directly verify whether certain domain knowledge, like conservation laws of energy and momentum, is satisfied or not. This makes it difficult to verify if a method is really capable of modeling certain dynamics and adhering to the corresponding constraints. To address this, we follow [8] to generate a synthetic dataset named $N$-body MNIST[3], which is an extension of MovingMNIST [50]. The dataset contains sequences of digits moving subject to the gravitational force from other digits. The governing equation for the motion is $\frac{d^2 \boldsymbol{x}_i}{dt^2} = -\sum_{j \neq i} \frac{Gm_j(\boldsymbol{x}_i - \boldsymbol{x}_j)}{(|\boldsymbol{x}_i - \boldsymbol{x}_j| + d_{\text{soft}})^r}$, where $\boldsymbol{x}_i$ is the spatial coordinates of the $i$-th digit, $G$ is the gravitational constant, $m_j$ is the mass of the $j$-th digit, $r$ is a constant representing the power scale in the gravitational law, and $d_{\text{soft}}$ is a small softening distance that ensures numerical stability. The motion occurs within a $64 \times 64$ frame. When a digit hits the boundaries of the frame, it bounces back by elastic collision. We use $N = 3$ for chaotic 3-body motion [35]. The forecasting task is to predict 10-step ahead future frames $x \in \mathbb{R}^{10 \times 64 \times 64 \times 1}$ given the length-10 context $y \in \mathbb{R}^{10 \times 64 \times 64 \times 1}$. We generate 20,000 sequences for training and 1,000 sequences for testing. Empirical studies on such a synthetic dataset with known dynamics helps provide useful insights for model development and evaluation.

**Evaluation.** In addition to standard metrics MSE, MAE and SSIM, we also report the scores of Fréchet Video Distance (FVD) [51], a metric for evaluating the visual quality of generated videos. Similar to Fréchet Inception Distance (FID) [20] for evaluating image generation, FVD estimates the distance between the learned distribution and the true data distribution by comparing the statistics of feature vectors extracted from the generations and the real data. The inception network used in FVD for feature extraction is pre-trained on video classification and is not specifically adapted for processing "unnatural videos" such as spatiotemporal observation data in Earth systems. Consequently, the FVD scores on the $N$-body MNIST and SEVIR datasets cannot be directly compared with those on natural video datasets. Nevertheless, the relative ranking of the FVD scores remains a meaningful indicator of model ability to achieve high visual quality, as FVD has shown consistency with expert evaluations across various domains beyond natural images [38, 26]. Scores for all involved metrics are calculated using an ensemble of eight samples from each model.

### A.0.1 Comparison with the State of the Art

We evaluate seven deterministic spatiotemporal forecasting models: **UNet** [55], **ConvLSTM** [47], **PredRNN** [61], **PhyDNet** [11], **E3D-LSTM** [60], **Rainformer** [1] and **Earthformer** [8], as well as two probabilistic spatiotemporal forecasting models: **VideoGPT** [65] and **LDM** [42]. All baselines are trained following the default configurations in their officially released code. More implementation details of baselines are provided in Appendix C.2. Results in Table 2 show that PreDiff outperforms these baselines by a large margin in both conventional video prediction metrics (i.e., MSE, MAE, SSIM), and a perceptual quality metric, FVD. The example predictions in Fig. 4 demonstrate that PreDiff generate predictions with sharp and clear digits in accurate positions. In contrast, deterministic baselines resort to generating blurry predictions to accommodate uncertainty. Probabilistic baselines, though producing sharp strokes, either predict *incorrect* positions or *fail to reconstruct* the digits. The performance gap between LDM [42] and PreDiff serves as an ablation study that highlights the importance of the latent backbone's spatiotemporal modeling capacity. Specifically, the Earthformer-UNet utilized in PreDiff demonstrates superior performance compared to the UNet in LDM [42].

### A.0.2 Knowledge Alignment: Energy Conservation

In the $N$-body MNIST simulation, digits move based on Newton's law of gravity, and interact with the boundaries through elastic collisions. Consequently, this system obeys the law of conservation of energy. The total energy of the whole system $E(x^j)$ at any future time step $j$ during evolution should equal the total energy at the last observation time step $E(y^{L_{\text{in}}})$.

---

[3]Code available at `https://github.com/amazon-science/earth-forecasting-transformer/tree/main/src/earthformer/datasets/nbody`

Figure 4: A set of example predictions on the $N$-body MNIST test set. From top to bottom: context sequence $y$, target sequence $x$, predictions by ConvLSTM [47], Earthformer [8], VideoGPT [65], LDM [42], PreDiff, and PreDiff with knowledge alignment (PreDiff-KA). E.MSE denotes the average error between the total energy (kinetic + potential) of the predictions $E(\widehat{x}^j)$ and the total energy of the last context frame $E(y^{L_{\text{in}}})$. The red dashed line is to help the reader to judge the position of the digit "2" in the last frame.

We impose the law of conservation of energy for the knowledge alignment on $N$-body MNIST in the form of (4) :

$$\mathcal{F}(\widehat{x}, y) \equiv [E(\widehat{x}^1), \ldots, E(\widehat{x}^{L_{\text{out}}})]^T, \tag{6}$$

$$\mathcal{F}_0(y) \equiv [E(y^{L_{\text{in}}}), \ldots, E(y^{L_{\text{in}}})]^T. \tag{7}$$

The ground-truth values of the total energy $E(y^{L_{\text{in}}})$ and $E(x^j)$ are directly accessible since $N$-body MNIST is a synthetic dataset from simulation. The total energy can be derived from the velocities (kinetic energy) and positions (potential energy) of the moving digits. A knowledge alignment network $U_\phi$ is trained following Alg. 1 to guide the PreDiff to generate forecasts $\widehat{x}$ that conserve the same energy as the initial step $E(y^{L_{\text{in}}})$.

To verify the effectiveness of the knowledge alignment on guiding the generations to comply with the law of conservation of energy, we train an energy detector $E_{\text{det}}(\widehat{x})^4$ that detects the total energy of the forecasts $\widehat{x}$. We evaluate the energy error between the forecasts and the initial energy using E.MSE$(\widehat{x}, y) \equiv \text{MSE}(E_{\text{det}}(\widehat{x}), E(y^{L_{\text{in}}}))$ and E.MAE$(\widehat{x}, y) \equiv \text{MAE}(E_{\text{det}}(\widehat{x}), E(y^{L_{\text{in}}}))$. In this evaluation, we exclude the methods that generate blurred predictions with ambiguous digit positions. We only focus on the methods that are capable of producing clear digits in precise positions.

As illustrated in Table 2, PreDiff-KA substantially outperforms all baseline methods and PreDiff without knowledge alignment in E.MSE and E.MAE. This demonstrates that the forecasts of PreDiff-KA comply much better with the law of conservation of energy, while still maintaining high visual quality with an FVD score of $4.063$.

---

[4] The test MSE of the energy detector is $5.56 \times 10^{-5}$, which is much smaller than the scores of E.MSE shown in Table 2. This indicates that the energy detector has high precision and reliability for verifying energy conservation in the model forecasts.

Table 2: Performance comparison on $N$-body MNIST. We report conventional frame quality metrics (MSE, MAE, SSIM), along with Fréchet Video Distance (FVD) [51] for assessing visual quality. Energy conservation is evaluated via E.MSE and E.MAE between the energy of predictions $E_{\mathrm{det}}(\widehat{x})$ and the initial energy $E(y^{L_{\mathrm{in}}})$. Lower values on the energy metrics indicate better compliance with conservation of energy.

| Model | #Param. (M) | Frame Metrics | | | | Energy Metrics | |
| --- | --- | --- | --- | --- | --- | --- | --- |
| | | MSE ↓ | MAE ↓ | SSIM ↑ | FVD ↓ | E.MSE ↓ | E.MAE ↓ |
| Target | - | 0.000 | 0.000 | 1.0000 | 0.000 | 0.0132 | 0.0697 |
| Persistence | - | 104.9 | 139.0 | 0.7270 | 168.3 | - | - |
| UNet [55] | 16.6 | 38.90 | 94.29 | 0.8260 | 142.3 | - | - |
| ConvLSTM [47] | 14.0 | 32.15 | 72.64 | 0.8886 | 86.31 | - | - |
| PredRNN [61] | 23.8 | 21.76 | 54.32 | 0.9288 | 20.65 | - | - |
| PhyDNet [11] | 3.1 | 28.97 | 78.66 | 0.8206 | 178.0 | - | - |
| E3D-LSTM [60] | 12.9 | 22.98 | 62.52 | 0.9131 | 22.28 | - | - |
| Rainformer [1] | 19.2 | 38.89 | 96.47 | 0.8036 | 163.5 | - | - |
| Earthformer [8] | 7.6 | 14.82 | 39.93 | 0.9538 | 6.798 | - | - |
| VideoGPT [65] | 92.2 | 53.68 | 77.42 | 0.8468 | 39.28 | 0.0228 | 0.1092 |
| LDM [42] | 410.3 | 46.29 | 72.19 | 0.8773 | 3.432 | 0.0243 | 0.1172 |
| PreDiff | 120.7 | **9.492** | **25.01** | **0.9716** | **0.987** | 0.0226 | 0.1083 |
| PreDiff-KA | 129.4 | 21.90 | 43.57 | 0.9303 | 4.063 | **0.0039** | **0.0443** |

Furthermore, we detect energy errors in the target data sequences. The first row of Table 2 indicates that even the target from the training data may not strictly adhere to the prior knowledge. This could be due to discretization errors in the simulation. Table 2 shows that all baseline methods and PreDiff have larger energy errors than the target, meaning purely data-oriented approaches cannot eliminate the impact of noise in the training data. In contrast, PreDiff-KA, guided by the law of conservation of energy, overcomes the intrinsic defects in the training data, achieving even lower energy errors compared to the target.

A typical example shown in Fig. 4 demonstrates that while PreDiff precisely reproduces the ground-truth position of digit "2" in the last frame (aligned to the red dashed line), resulting in nearly the same energy error (E.MSE = 0.0277) as the ground-truth's (E.MSE = 0.0261), PreDiff-KA successfully corrects the motion of digit "2", providing it with physically plausible velocity and position (slightly off the red dashed line). The knowledge alignment ensures that the generation complies better with the law of conservation of energy, resulting in a much lower E.MSE = 0.0086. On the contrary, none of the evaluated baselines can overcome the intrinsic noise from the data, resulting in energy errors comparable to or larger than that of the ground-truth.

Notice that the pixel-wise scores MSE, MAE and SSIM are less meaningful for evaluating PreDiff-KA, since correcting the noise of the energy results in changing the velocities and positions of the digits. A minor change in the position of a digit can cause a large pixel-wise error, even though the digit is still generated sharply and in high quality as shown in Fig. 4.

# B  Related Work

**Deep learning for precipitation nowcasting**   In recent years, the field of DL has experienced remarkable advancements, revolutionizing various domains of study, including Earth science. One area where DL has particularly made significant strides is in the field of Earth system forecasting, especially precipitation nowcasting. Precipitation nowcasting benefits from the success of DL architectures such as convolutional neural networks (CNNs), recurrent neural networks (RNNs), and Transformers, which have demonstrated their effectiveness in handling spatiotemporal tensors, the typical formulation for Earth system observation data. ConvLSTM [47], a pioneering approach in DL for precipitation nowcasting, combines the strengths of CNNs and LSTMs processing spatial and temporal data. PredRNN [61] builds upon ConvLSTM by incorporating a spatiotemporal memory flow structure. E3D-LSTM [60] integrates 3D CNN to LSTM to enhance long-term high-level relation modeling. PhyDNet [11] incorporated partial differential equation (PDE) constraints in the latent space. MetNet [49] and its successor, MetNet-2 [5], propose architectures based on ConvLSTM and dilated CNN, enabling skillful precipitation forecasts up to twelve hours ahead. DGMR [41] takes an adversarial training approach to generate sharp and accurate nowcasts, addressing the issue of blurry predictions.

In addition to precipitation nowcasting, there has been a surge in the modeling of global weather and medium-range weather forecasting due to the availability of extensive Earth observation data, such as the European Centre for Medium-Range Weather Forecasts (ECMWF)'s ERA5 [19] dataset. Several DL-based models have emerged in this area. FourCastNet [37] proposes an architecture with Adaptive Fourier Neural Operators (AFNO) [12] as building blocks for autoregressive weather forecasting. FengWu [3] introduces a multi-model Transformer-based global medium-range weather forecast model that achieves skillful forecasts up to ten days ahead. GraphCast [29] combines graph neural networks with convolutional LSTMs to tackle sub-seasonal forecasting tasks, representing weather phenomena as spatiotemporal graphs. Pangu-Weather [2] proposes a 3D Transformer model with Earth-specific priors and a hierarchical temporal aggregation strategy for medium-range global weather forecasting. While recent years have seen remarkable progress in DL for precipitation nowcasting, existing methods still face some limitations. Some methods are deterministic, failing to capture uncertainty and resulting in blurry generation. Others lack the capability of incorporating prior knowledge, which is crucial for machine learning for science. In contrast, PreDiff captures the uncertainty in the underlying data distribution via diffusion models, avoiding simply averaging all possibilities into blurry forecasts. Our knowledge alignment mechanism facilitates post-training alignment with physical principles and domain-specific prior knowledge.

**Diffusion models**   Diffusion models (DMs) [22] are a class of generative models that have become increasingly popular in recent years. DMs learn the data distribution by constructing a forward process that adds noise to the data, and then approximating the reverse process to remove the noise. Latent diffusion models (LDMs) [42] are a variant of DMs that are trained on latent vector outputs from a variational autoencoder. LDMs have been shown to be more efficient in both training and inference compared to original DMs. Building on the success of DMs in image generation, DMs have also been adopted for video generation. MCVD [56] trains a DM by randomly masking past and/or future frames in blocks and conditioning on the remaining frames. It generates long videos by autoregressively sampling blocks of frames in a sliding window manner. PVDM [66] projects videos into low-dimensional latent space as 2D vectors, and presents a joint training of unconditional and frame conditional video generations. LFDM [36] employs a flow predictor to estimate latent flows between video frames and learns a DM for temporal latent flow generation. VideoFusion [32] decomposes the transition noise in DMs into per-frame noise and the noise along time axis, and trains two networks jointly to match the noise decomposition. While DMs have demonstrated impressive performance in video synthesis, its applications to precipitation nowcasting and other Earth science tasks have not been well explored. Hatanaka et al. [16] uses DMs to super-resolve coarse numerical predictions for solar forecast. Concurrent to our work, LDCast [30] applies LDMs for precipitation nowcasting. However, LDCast has not studied how to integrate prior knowledge to the DM, which is a unique advantage and novelty of PreDiff.

**Conditional controls on diffusion models**   Another key advantage of DMs is the ability to condition generation on text, class labels, and other modalities for controllable and diverse output. For instance, ControlNet [68] enables fine-tuning a pretrained DM by freezing the base model and training a copy

end-to-end with conditional inputs. Composer [24] decomposes images into representative factors used as conditions to guide the generation. Beyond text and class labels, conditions in other modalities, including physical constraints, can also be leveraged to provide valuable guidance. TopDiff [33] constrains topology optimization using load, boundary conditions, and volume fraction. Physdiff [67] trains a physics-based motion projection module with reinforcement learning to project denoised motions in diffusion steps into physically plausible ones. Nonetheless, while conditional control has proven to be a powerful technique in various domains, its application in DL for precipitation nowcasting remains an unexplored area.

# C Implementation Details

All experiments are conducted on machines with NVIDIA A10G GPUs (24GB memoery). All models, including PreDiff, knowledge alignment networks and the baselines, can fit in a single GPU without the need for gradient checkpointing or model parallelization.

## C.1 PreDiff

**Frame-wise autoencoder** We follow [7, 42] to build frame-wise VAEs (not VQVAEs) and train them adversarially from scratch on $N$-body MNIST and SEVIR frames. As shown in Sec. 2.2, on $N$-body MNIST dataset, the spatial downsampling ratio is $4 \times 4$. A frame $x^j \in \mathbb{R}^{64 \times 64 \times 1}$ is encoded to $z^j \in \mathbb{R}^{16 \times 16 \times 3}$ by parameterizing $p_{\mathcal{E}}(z^j|x^j) = \mathcal{N}(\mu_{\mathcal{E}}(x^j)|\sigma_{\mathcal{E}}(x^j))$. On SEVIR dataset, the spatial downsampling ratio is $8 \times 8$. A frame $x^j \in \mathbb{R}^{128 \times 128 \times 1}$ is encoded to $z^j \in \mathbb{R}^{16 \times 16 \times 4}$ similarly.

The detailed configurations of the encoder and decoder of the VAE on $N$-body MNIST are shown in Table 3 and Table 4. The detailed configurations of the encoder and decoder of the VAE on SEVIR are shown in Table 5 and Table 6. The discriminators for adversarial training on $N$-body MNIST and SEVIR datasets share the same configurations, which are shown in Table 7.

**Latent diffusion model that instantiates** $p_\theta(z_{t-1}|z_t, z_{\text{cond}})$ Stemming from Earthformer [8], we build *Earthformer-UNet*, which is a hierarchical UNet with self cuboid attention [8] layers as basic building blocks, as shown in Fig. 5.

On $N$-body MNIST, it takes the concatenation along the temporal dimension (the sequence length axis) of $z_{\text{cond}} \in \mathbb{R}^{10 \times 16 \times 16 \times 3}$ and $z_t \in \mathbb{R}^{10 \times 16 \times 16 \times 3}$ as input, and outputs $z_{t-1} \in \mathbb{R}^{10 \times 16 \times 16 \times 3}$. On SEVIR, it takes the concatenation along the temporal dimension (the sequence length axis) of $z_{\text{cond}} \in \mathbb{R}^{7 \times 16 \times 16 \times 4}$ and $z_t \in \mathbb{R}^{6 \times 16 \times 16 \times 4}$ as input, and outputs $z_{t-1} \in \mathbb{R}^{6 \times 16 \times 16 \times 4}$. Besides, we add the embedding of the denoising step $t$ to the state in front of each cuboid attention block via an embeding layer `TEmbed`, following [22]. The detailed configurations of the Earthformer-UNet is described in Table 8.

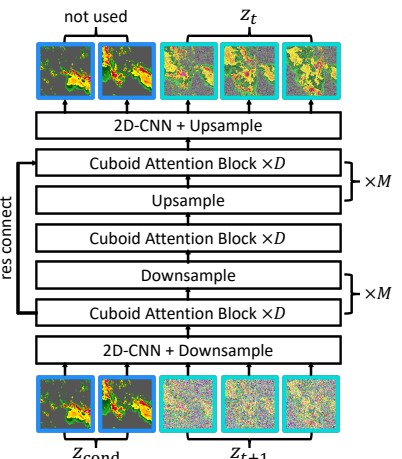

Figure 5: **Earthformer-UNet architecture.** PreDiff employs an Earthformer-UNet as the backbone for parameterizing the latent diffusion model $p_\theta(z_t|z_{t+1}, z_{\text{cond}})$. It takes the concatenation of the latent context $z_{\text{cond}}$ (in the blue border) and the previous-step noisy latent future $z_{t+1}$ (in the cyan border) along the temporal dimension (the sequence length axis) as input, and outputs $z_t$. (Best viewed in color).

**Knowledge alignment networks** A knowledge alignment network parameterizes $U_\phi(z_t, t, y)$ to predict $\mathcal{F}(\widehat{x}, y)$ using the noisy latent $z_t$. In practice, we build an Earthformer encoder [8] with a final pooling block as the knowledge alignment network to parameterize $U_\phi(z_t, t, z_{\text{cond}})$, which takes $t$, and the concatenation of $z_{\text{cond}}$ and $z_t$, instead of $t$, $y$ and $z_t$ as the inputs. We find this implementation accurate enough when $t$ is small. The detailed configurations of the knowledge alignment network is described in Table 9

**Optimization** We train the frame-wise VAEs using the Adam optimizer [27] following [7]. We train the latent Earthformer-UNet and the knowledge alignment network using the AdamW optimizer [31] following [8]. Detailed configurations are shown in Table 10, Table 11 and Table 12 for the frame-wise VAE, the latent Earthformer-UNet and the knowledge alignment network, respectively. We adopt data parallel and gradient accumulation to use a larger total batch size while the GPU can only afford a smaller micro batch size.

Table 3: The details of the encoder of the frame-wise VAE on $N$-body MNIST frames. It encodes an input frame $x^j \in \mathbb{R}^{64 \times 64 \times 1}$ into a latent $z^j \in \mathbb{R}^{16 \times 16 \times 3}$. Conv3 $\times$ 3 is the 2D convolutional layer with $3 \times 3$ kernel. GroupNorm32 is the Group Normalization (GN) layer [64] with 32 groups. SiLU is the Sigmoid Linear Unit activation layer [18] with function $\texttt{SiLU}(x) = x \cdot \texttt{sigmoid}(x)$. The Attention is the self attention layer [54] that first maps the input to queries $Q$, keys $K$ and values $V$ by three Linear layers, and then does self attention operation: $\texttt{Attention}(x) = \texttt{Softmax}(QK^T/\sqrt{C})V$.

| Block | Layer | Resolution | Channels |
|---|---|---|---|
| Input $x^j$ | - | $64 \times 64$ | 1 |
| 2D CNN | Conv3 $\times$ 3 | $64 \times 64$ | $1 \rightarrow 128$ |
| | GroupNorm32 | $64 \times 64$ | 128 |
| | Conv3 $\times$ 3 | $64 \times 64$ | 128 |
| ResNet Block $\times 2$ | GroupNorm32 | $64 \times 64$ | 128 |
| | Conv3 $\times$ 3 | $64 \times 64$ | 128 |
| | SiLU | $64 \times 64$ | 128 |
| Downsampler | Conv3 $\times$ 3 | $64 \times 64 \rightarrow 32 \times 32$ | 128 |
| | GroupNorm32 | $32 \times 32$ | 128 |
| | Conv3 $\times$ 3 | $32 \times 32$ | $128 \rightarrow 256, 256$ |
| ResNet Block $\times 2$ | GroupNorm32 | $32 \times 32$ | 256 |
| | Conv3 $\times$ 3 | $32 \times 32$ | 256 |
| | SiLU | $32 \times 32$ | 256 |
| Downsampler | Conv3 $\times$ 3 | $32 \times 32 \rightarrow 16 \times 16$ | 256 |
| | GroupNorm32 | $16 \times 16$ | 256 |
| | Conv3 $\times$ 3 | $16 \times 16$ | $256 \rightarrow 512, 512$ |
| ResNet Block $\times 2$ | GroupNorm32 | $16 \times 16$ | 512 |
| | Conv3 $\times$ 3 | $16 \times 16$ | 512 |
| | SiLU | $16 \times 16$ | 512 |
| | GroupNorm32 | $16 \times 16$ | 512 |
| Self Attention Block | Attention | $16 \times 16$ | 512 |
| | Linear | $16 \times 16$ | 512 |
| | GroupNorm32 | $16 \times 16$ | 512 |
| | Conv3 $\times$ 3 | $16 \times 16$ | 512 |
| ResNet Block $\times 2$ | GroupNorm32 | $16 \times 16$ | 512 |
| | Conv3 $\times$ 3 | $16 \times 16$ | 512 |
| | SiLU | $16 \times 16$ | 512 |
| | GroupNorm32 | $16 \times 16$ | 512 |
| | SiLU | $16 \times 16$ | 512 |
| Output Block | Conv3 $\times$ 3 | $16 \times 16$ | $512 \rightarrow 6$ |
| | Conv3 $\times$ 3 | $16 \times 16$ | 6 |

Table 4: The details of the decoder of the frame-wise VAE on $N$-body MNIST frames. It decodes a latent $z^j \in \mathbb{R}^{16 \times 16 \times 3}$ back to a frame in pixel space $x^j \in \mathbb{R}^{64 \times 64 \times 1}$. $\texttt{Conv3} \times 3$ is the 2D convolutional layer with $3 \times 3$ kernel. $\texttt{GroupNorm32}$ is the Group Normalization (GN) layer [64] with 32 groups. $\texttt{SiLU}$ is the Sigmoid Linear Unit activation layer [18] with function $\texttt{SiLU}(x) = x \cdot \texttt{sigmoid}(x)$. The $\texttt{Attention}$ is the self attention layer [54] that first maps the input to queries $Q$, keys $K$ and values $V$ by three $\texttt{Linear}$ layers, and then does self attention operation: $\texttt{Attention}(x) = \texttt{Softmax}(QK^T/\sqrt{C})V$.

| Block | Layer | Resolution | Channels |
|---|---|---|---|
| Input $z^j$ | - | $16 \times 16$ | 3 |
| 2D CNN | $\texttt{Conv3} \times 3$ | $16 \times 16$ | 3 |
| | $\texttt{Conv3} \times 3$ | $16 \times 16$ | $3 \to 512$ |
| Self Attention Block | $\texttt{GroupNorm32}$ | $16 \times 16$ | 512 |
| | $\texttt{Attention}$ | $16 \times 16$ | 512 |
| | $\texttt{Linear}$ | $16 \times 16$ | 512 |
| ResNet Block $\times 3$ | $\texttt{GroupNorm32}$ | $16 \times 16$ | 512 |
| | $\texttt{Conv3} \times 3$ | $16 \times 16$ | 512 |
| | $\texttt{GroupNorm32}$ | $16 \times 16$ | 512 |
| | $\texttt{Conv3} \times 3$ | $16 \times 16$ | 512 |
| | $\texttt{SiLU}$ | $16 \times 16$ | 512 |
| Upsampler | $\texttt{Conv3} \times 3$ | $16 \times 16 \to 32 \times 32$ | 512 |
| ResNet Block $\times 3$ | $\texttt{GroupNorm32}$ | $32 \times 32$ | 512 |
| | $\texttt{Conv3} \times 3$ | $32 \times 32$ | $512 \to 256, 256, 256$ |
| | $\texttt{GroupNorm32}$ | $32 \times 32$ | 256 |
| | $\texttt{Conv3} \times 3$ | $32 \times 32$ | 256 |
| | $\texttt{SiLU}$ | $32 \times 32$ | 256 |
| Upsampler | $\texttt{Conv3} \times 3$ | $32 \times 32 \to 64 \times 64$ | 256 |
| ResNet Block $\times 3$ | $\texttt{GroupNorm32}$ | $64 \times 64$ | 256 |
| | $\texttt{Conv3} \times 3$ | $64 \times 64$ | $256 \to 128, 128, 128$ |
| | $\texttt{GroupNorm32}$ | $64 \times 64$ | 128 |
| | $\texttt{Conv3} \times 3$ | $64 \times 64$ | 128 |
| | $\texttt{SiLU}$ | $64 \times 64$ | 128 |
| Output Block | $\texttt{GroupNorm32}$ | $64 \times 64$ | 128 |
| | $\texttt{SiLU}$ | $64 \times 64$ | 128 |
| | $\texttt{Conv3} \times 3$ | $64 \times 64$ | $128 \to 1$ |

Table 5: The details of the encoder of the frame-wise VAE on SEVIR frames. It encodes an input frame $x^j \in \mathbb{R}^{128 \times 128 \times 1}$ into a latent $z^j \in \mathbb{R}^{16 \times 16 \times 4}$. Conv3 $\times$ 3 is the 2D convolutional layer with $3 \times 3$ kernel. GroupNorm32 is the Group Normalization (GN) layer [64] with 32 groups. SiLU is the Sigmoid Linear Unit activation layer [18] with function $\mathtt{SiLU}(x) = x \cdot \mathtt{sigmoid}(x)$. The Attention is the self attention layer [54] that first maps the input to queries $Q$, keys $K$ and values $V$ by three Linear layers, and then does self attention operation: $\mathtt{Attention}(x) = \mathtt{Softmax}(QK^T/\sqrt{C})V$.

| Block | Layer | Resolution | Channels |
|-------|-------|------------|----------|
| Input $x^j$ | - | $128 \times 128$ | 1 |
| 2D CNN | Conv3 $\times$ 3 | $128 \times 128$ | $1 \to 128$ |
| ResNet Block $\times 2$ | GroupNorm32 | $128 \times 128$ | 128 |
| | Conv3 $\times$ 3 | $128 \times 128$ | 128 |
| | GroupNorm32 | $128 \times 128$ | 128 |
| | Conv3 $\times$ 3 | $128 \times 128$ | 128 |
| | SiLU | $128 \times 128$ | 128 |
| Downsampler | Conv3 $\times$ 3 | $128 \times 128 \to 64 \times 64$ | 128 |
| ResNet Block $\times 2$ | GroupNorm32 | $64 \times 64$ | 128 |
| | Conv3 $\times$ 3 | $64 \times 64$ | $128 \to 256, 256$ |
| | GroupNorm32 | $64 \times 64$ | 256 |
| | Conv3 $\times$ 3 | $64 \times 64$ | 256 |
| | SiLU | $64 \times 64$ | 256 |
| Downsampler | Conv3 $\times$ 3 | $64 \times 64 \to 32 \times 32$ | 256 |
| ResNet Block $\times 2$ | GroupNorm32 | $32 \times 32$ | 256 |
| | Conv3 $\times$ 3 | $32 \times 32$ | $256 \to 512, 512$ |
| | GroupNorm32 | $32 \times 32$ | 512 |
| | Conv3 $\times$ 3 | $32 \times 32$ | 512 |
| | SiLU | $32 \times 32$ | 512 |
| Downsampler | Conv3 $\times$ 3 | $32 \times 32 \to 16 \times 16$ | 512 |
| ResNet Block $\times 2$ | GroupNorm32 | $16 \times 16$ | 512 |
| | Conv3 $\times$ 3 | $16 \times 16$ | 512 |
| | GroupNorm32 | $16 \times 16$ | 512 |
| | Conv3 $\times$ 3 | $16 \times 16$ | 512 |
| | SiLU | $16 \times 16$ | 512 |
| Self Attention Block | GroupNorm32 | $16 \times 16$ | 512 |
| | Attention | $16 \times 16$ | 512 |
| | Linear | $16 \times 16$ | 512 |
| ResNet Block $\times 2$ | GroupNorm32 | $16 \times 16$ | 512 |
| | Conv3 $\times$ 3 | $16 \times 16$ | 512 |
| | GroupNorm32 | $16 \times 16$ | 512 |
| | Conv3 $\times$ 3 | $16 \times 16$ | 512 |
| | SiLU | $16 \times 16$ | 512 |
| Output Block | GroupNorm32 | $16 \times 16$ | 512 |
| | SiLU | $16 \times 16$ | 512 |
| | Conv3 $\times$ 3 | $16 \times 16$ | $512 \to 8$ |
| | Conv3 $\times$ 3 | $16 \times 16$ | 8 |

Table 6: The details of the decoder of the frame-wise VAE on SEVIR frames. It decodes a latent $z^j \in \mathbb{R}^{16 \times 16 \times 4}$ back to a frame in pixel space $x^j \in \mathbb{R}^{128 \times 128 \times 1}$. Conv3 $\times$ 3 is the 2D convolutional layer with $3 \times 3$ kernel. GroupNorm32 is the Group Normalization (GN) layer [64] with 32 groups. SiLU is the Sigmoid Linear Unit activation layer [18] with function $\texttt{SiLU}(x) = x \cdot \texttt{sigmoid}(x)$. The Attention is the self attention layer [54] that first maps the input to queries $Q$, keys $K$ and values $V$ by three Linear layers, and then does self attention operation: $\texttt{Attention}(x) = \texttt{Softmax}(QK^T/\sqrt{C})V)$.

| Block | Layer | Resolution | Channels |
|---|---|---|---|
| Input $z^j$ | - | $16 \times 16$ | 4 |
| 2D CNN | Conv3 $\times$ 3 | $16 \times 16$ | 4 |
| | Conv3 $\times$ 3 | $16 \times 16$ | $4 \to 512$ |
| Self Attention Block | GroupNorm32 | $16 \times 16$ | 512 |
| | Attention | $16 \times 16$ | 512 |
| | Linear | $16 \times 16$ | 512 |
| ResNet Block $\times$3 | GroupNorm32 | $16 \times 16$ | 512 |
| | Conv3 $\times$ 3 | $16 \times 16$ | 512 |
| | GroupNorm32 | $16 \times 16$ | 512 |
| | Conv3 $\times$ 3 | $16 \times 16$ | 512 |
| | SiLU | $16 \times 16$ | 512 |
| Upsampler | Conv3 $\times$ 3 | $16 \times 16 \to 32 \times 32$ | 512 |
| ResNet Block $\times$3 | GroupNorm32 | $32 \times 32$ | 512 |
| | Conv3 $\times$ 3 | $32 \times 32$ | 512 |
| | GroupNorm32 | $32 \times 32$ | 512 |
| | Conv3 $\times$ 3 | $32 \times 32$ | 512 |
| | SiLU | $32 \times 32$ | 512 |
| Upsampler | Conv3 $\times$ 3 | $32 \times 32 \to 64 \times 64$ | 512 |
| ResNet Block $\times$3 | GroupNorm32 | $64 \times 64$ | 512 |
| | Conv3 $\times$ 3 | $64 \times 64$ | $512 \to 256, 256, 256$ |
| | GroupNorm32 | $64 \times 64$ | 256 |
| | Conv3 $\times$ 3 | $64 \times 64$ | 256 |
| | SiLU | $64 \times 64$ | 256 |
| Upsampler | Conv3 $\times$ 3 | $64 \times 64 \to 128 \times 128$ | 256 |
| ResNet Block $\times$3 | GroupNorm32 | $128 \times 128$ | 256 |
| | Conv3 $\times$ 3 | $128 \times 128$ | $256 \to 128, 128, 128$ |
| | GroupNorm32 | $128 \times 128$ | 128 |
| | Conv3 $\times$ 3 | $128 \times 128$ | 128 |
| | SiLU | $128 \times 128$ | 128 |
| Output Block | GroupNorm32 | $128 \times 128$ | 128 |
| | SiLU | $128 \times 128$ | 128 |
| | Conv3 $\times$ 3 | $128 \times 128$ | $128 \to 1$ |

Table 7: The details of the discriminator for the adversarial loss of on $N$-body MNIST and SEVIR frames. `Conv4 × 4` is the 2D convolutional layer with $4 \times 4$ kernel, $2 \times 2$ or $1 \times 1$ stride, and $1 \times 1$ padding. `BatchNorm` is the Batch Normalization (BN) layer [25] . The negative slope in `LeakyReLU` is $0.2$.

| Block | Layer | Resolution | | Channels |
| | | $N$-body MNIST | SEVIR | |
| --- | --- | --- | --- | --- |
| Input $x^j$ | - | $64 \times 64$ | $128 \times 128$ | 1 |
| 2D CNN | Conv4 × 4 | $64 \times 64 \rightarrow 32 \times 32$ | $128 \times 128 \rightarrow 64 \times 64$ | $1 \rightarrow 64$ |
| | LeakyReLU | $32 \times 32$ | $64 \times 64$ | 64 |
| Downsampler | Conv4 × 4 | $32 \times 32 \rightarrow 16 \times 16$ | $64 \times 64 \rightarrow 32 \times 32$ | $64 \rightarrow 128$ |
| | BatchNorm | $16 \times 16$ | $32 \times 32$ | 128 |
| | LeakyReLU | $16 \times 16$ | $32 \times 32$ | 128 |
| Downsampler | Conv4 × 4 | $16 \times 16 \rightarrow 8 \times 8$ | $32 \times 32 \rightarrow 16 \times 16$ | $128 \rightarrow 256$ |
| | BatchNorm | $8 \times 8$ | $16 \times 16$ | 256 |
| | LeakyReLU | $8 \times 8$ | $16 \times 16$ | 256 |
| Downsampler | Conv4 × 4 | $8 \times 8 \rightarrow 7 \times 7$ | $16 \times 16 \rightarrow 15 \times 15$ | $256 \rightarrow 512$ |
| | BatchNorm | $7 \times 7$ | $15 \times 15$ | 512 |
| | LeakyReLU | $7 \times 7$ | $15 \times 15$ | 512 |
| Output Block | Conv4 × 4 | $7 \times 7 \rightarrow 6 \times 6$ | $15 \times 15 \rightarrow 14 \times 14$ | 1 |
| | AvgPool | $6 \times 6 \rightarrow 1$ | $15 \times 15 \rightarrow 1$ | 1 |

Table 8: The details of the Earthformer-UNet as the latent diffusion backbone on $N$-body MNIST and SEVIR datasets. The `ConcatMask` layer for the Observation Mask block concatenates one more channel to the input to indicates whether the input is the encoded observation $z_{\text{cond}}$ or the noisy latent $z_t$. 1 for $z_{\text{cond}}$ and 0 for $z_t$. `Conv3 × 3` is the 2D convolutional layer with $3 \times 3$ kernel. `GroupNorm32` is the Group Normalization (GN) layer [64] with 32 groups. If the number of the input data channels is smaller than 32, then the number of groups is set to the number of channels. `SiLU` is the Sigmoid Linear Unit activation layer [18] with function $\texttt{SiLU}(x) = x \cdot \texttt{sigmoid}(x)$. The negative slope in `LeakyReLU` is 0.1. `Dropout` is the dropout layer [21] with the probability 0.1 to drop an element to be zeroed. The `FFN` consists of two `Linear` layers separated by a `GeLU` activation layer [18]. `PosEmbed` is the positional embedding layer [54] that adds learned positional embeddings to the input. `TEmbed` is the embedding layer [22] that embeds the denoising step $t$. `PatchMerge` splits a 2D input tensor with $C$ channels into $N$ non-overlapping $p \times p$ patches and merges the spatial dimensions into channels, gets $N$ $1 \times 1$ patches with $p^2 \cdot C$ channels and concatenates them back along spatial dimensions. Residual connections [17] are added from blocks in the downsampling phase to corresponding blocks in the upsampling phase.

| Block | Layer | Spatial Resolution | Channels $N$-body MNIST | SEVIR |
|---|---|---|---|---|
| Input $[z_{\text{cond}}, z_t]$ | - | $16 \times 16$ | 3 | 4 |
| Observation Mask | ConcatMask | $16 \times 16$ | $3 \to 4$ | $4 \to 5$ |
| Projector | GroupNorm32 | $16 \times 16$ | 4 | 5 |
| | SiLU | $16 \times 16$ | 4 | 5 |
| | Conv3 × 3 | $16 \times 16$ | $4 \to 256$ | $5 \to 256$ |
| | GroupNorm32 | $16 \times 16$ | 256 | |
| | SiLU | $16 \times 16$ | 256 | |
| | Dropout | $16 \times 16$ | 256 | |
| | Conv3 × 3 | $16 \times 16$ | 256 | |
| Positional Embedding | PosEmbed | $16 \times 16$ | 256 | |
| Cuboid Attention Block ×4 | TEmbed | $16 \times 16$ | 256 | |
| | LayerNorm | $16 \times 16$ | 256 | |
| | Cuboid(T, 1, 1) | $16 \times 16$ | 256 | |
| | FFN | $16 \times 16$ | 256 | |
| | LayerNorm | $16 \times 16$ | 256 | |
| | Cuboid(1, H, 1) | $16 \times 16$ | 256 | |
| | FFN | $16 \times 16$ | 256 | |
| | LayerNorm | $16 \times 16$ | 256 | |
| | Cuboid(1, 1, W) | $16 \times 16$ | 256 | |
| | FFN | $16 \times 16$ | 256 | |
| Downsampler | PatchMerge | $16 \times 16 \to 8 \times 8$ | $256 \to 1024$ | |
| | LayerNorm | $8 \times 8$ | 1024 | |
| | Linear | $8 \times 8$ | 1024 | |
| Cuboid Attention Block ×8 | TEmbed | $8 \times 8$ | 1024 | |
| | LayerNorm | $8 \times 8$ | 1024 | |
| | Cuboid(T, 1, 1) | $8 \times 8$ | 1024 | |
| | FFN | $8 \times 8$ | 1024 | |
| | LayerNorm | $8 \times 8$ | 1024 | |
| | Cuboid(1, H, 1) | $8 \times 8$ | 1024 | |
| | FFN | $8 \times 8$ | 1024 | |
| | LayerNorm | $8 \times 8$ | 1024 | |
| | Cuboid(1, 1, W) | $8 \times 8$ | 1024 | |
| | FFN | $8 \times 8$ | 1024 | |
| Upsampler | NearestNeighborInterp | $8 \times 8 \to 16 \times 16$ | 1024 | |
| | Conv3 × 3 | $16 \times 16$ | $1024 \to 256$ | |
| Cuboid Attention Block ×4 | TEmbed | $16 \times 16$ | 256 | |
| | LayerNorm | $16 \times 16$ | 256 | |
| | Cuboid(T, 1, 1) | $16 \times 16$ | 256 | |
| | FFN | $16 \times 16$ | 256 | |
| | LayerNorm | $16 \times 16$ | 256 | |
| | Cuboid(1, H, 1) | $16 \times 16$ | 256 | |
| | FFN | $16 \times 16$ | 256 | |
| | LayerNorm | $16 \times 16$ | 256 | |
| | Cuboid(1, 1, W) | $16 \times 16$ | 256 | |
| | FFN | $16 \times 16$ | 256 | |
| Output Block | Linear | $16 \times 16$ | $256 \to 3$ | $256 \to 4$ |

Table 9: The details of the Earthformer encoders for the parameterization of the knowledge alignment networks $U_\phi(z_t, t, z_{\text{cond}})$ on $N$-body MNIST and SEVIR datasets. The `ConcatMask` layer for the Observation Mask block concatenates one more channel to the input to indicates whether the input is the encoded observation $z_{\text{cond}}$ or the noisy latent $z_t$. 1 for $z_{\text{cond}}$ and 0 for $z_t$. `Conv3 × 3` is the 2D convolutional layer with $3 \times 3$ kernel. `GroupNorm32` is the Group Normalization (GN) layer [64] with 32 groups. If the number of the input data channels is smaller than 32, then the number of groups is set to the number of channels. `SiLU` is the Sigmoid Linear Unit activation layer [18] with function $\texttt{SiLU}(x) = x \cdot \texttt{sigmoid}(x)$. The negative slope in `LeakyReLU` is 0.1. `Dropout` is the dropout layer [21] with the probability 0.1 to drop an element to be zeroed. The `FFN` consists of two `Linear` layers separated by a GeLU activation layer [18]. `PosEmbed` is the positional embedding layer [54] that adds learned positional embeddings to the input. `TEmbed` is the embedding layer [22] that embeds the denoising step $t$. `PatchMerge` splits a 2D input tensor with $C$ channels into $N$ non-overlapping $p \times p$ patches and merges the spatial dimensions into channels, gets $N$ $1 \times 1$ patches with $p^2 \cdot C$ channels and concatenates them back along spatial dimensions. Residual connections [17] are added from blocks in the downsampling phase to corresponding blocks in the upsampling phase. The `Attention` is the self attention layer [54] with an extra "cls" token for information aggregation. It first flattens the input and concatenates it with the "cls" token. Then it maps the concatenated input to queries $Q$, keys $K$ and values $V$ by three `Linear` layers, and then does self attention operation: $\texttt{Attention}(x) = \texttt{Softmax}(QK^T/\sqrt{C})V$. Finally, the value of the "cls" token after self attention operation serves as the layer's output.

| Block | Layer | Spatial Resolution | Channels $N$-body MNIST | SEVIR |
|---|---|---|---|---|
| Input $[z_{\text{cond}}, z_t]$ | - | $16 \times 16$ | 3 | 4 |
| Observation Mask | ConcatMask | $16 \times 16$ | $3 \to 4$ | $4 \to 5$ |
| Projector | GroupNorm32 | $16 \times 16$ | 4 | 5 |
| | SiLU | $16 \times 16$ | 4 | 5 |
| | Conv3 × 3 | $16 \times 16$ | $4 \to 64$ | $5 \to 64$ |
| | GroupNorm32 | $16 \times 16$ | 64 | |
| | SiLU | $16 \times 16$ | 64 | |
| | Dropout | $16 \times 16$ | 64 | |
| | Conv3 × 3 | $16 \times 16$ | 64 | |
| Positional Embedding | PosEmbed | $16 \times 16$ | 64 | |
| Cuboid Attention Block | TEmbed | $16 \times 16$ | 64 | |
| | LayerNorm | $16 \times 16$ | 64 | |
| | Cuboid(T, 1, 1) | $16 \times 16$ | 64 | |
| | FFN | $16 \times 16$ | 64 | |
| | LayerNorm | $16 \times 16$ | 64 | |
| | Cuboid(1, H, 1) | $16 \times 16$ | 64 | |
| | FFN | $16 \times 16$ | 64 | |
| | LayerNorm | $16 \times 16$ | 64 | |
| | Cuboid(1, 1, W) | $16 \times 16$ | 64 | |
| | FFN | $16 \times 16$ | 64 | |
| Downsampler | PatchMerge | $16 \times 16 \to 8 \times 8$ | $64 \to 256$ | |
| | LayerNorm | $8 \times 8$ | 256 | |
| | Linear | $8 \times 8$ | 256 | |
| Cuboid Attention Block | TEmbed | $8 \times 8$ | 256 | |
| | LayerNorm | $8 \times 8$ | 256 | |
| | Cuboid(T, 1, 1) | $8 \times 8$ | 256 | |
| | FFN | $8 \times 8$ | 256 | |
| | LayerNorm | $8 \times 8$ | 256 | |
| | Cuboid(1, H, 1) | $8 \times 8$ | 256 | |
| | FFN | $8 \times 8$ | 256 | |
| | LayerNorm | $8 \times 8$ | 256 | |
| | Cuboid(1, 1, W) | $8 \times 8$ | 256 | |
| | FFN | $8 \times 8$ | 256 | |
| Output Pooling Block | GroupNorm32 | $8 \times 8$ | 256 | |
| | Attention | $8 \times 8 \to 1$ | 256 | |
| | Linear | 1 | $256 \to 1$ | |

Table 10: Hyperparameters of the Adam optimizer for training frame-wise VAEs and discriminators on $N$-body MNIST and SEVIR datasets.

| Hyper-parameter of VAE | Value |
|---|---|
| Learning rate | $4.5 \times 10^{-6}$ |
| $\beta_1$ | 0.5 |
| $\beta_2$ | 0.9 |
| Weight decay | $10^{-2}$ |
| Batch size | 512 |
| Training epochs | 200 |
| Hyper-parameter of discriminator | Value |
| Learning rate | $4.5 \times 10^{-6}$ |
| $\beta_1$ | 0.5 |
| $\beta_2$ | 0.9 |
| Weight decay | $10^{-2}$ |
| Batch size | 512 |
| Training epochs | 200 |
| Training start step | 50000 |

Table 11: Hyperparameters of the AdamW optimizer for training LDMs on $N$-body MNIST and SEVIR datasets.

| Hyper-parameter of VAE | Value |
|---|---|
| Learning rate | $1.0 \times 10^{-3}$ |
| $\beta_1$ | 0.9 |
| $\beta_2$ | 0.999 |
| Weight decay | $10^{-5}$ |
| Batch size | 64 |
| Training epochs | 1000 |
| Warm up percentage | 10% |
| Learning rate decay | Cosine |

Table 12: Hyperparameters of the AdamW optimizer for training knowledge alignment networks on $N$-body MNIST and SEVIR datasets.

| Hyper-parameter of VAE | Value |
|---|---|
| Learning rate | $1.0 \times 10^{-3}$ |
| $\beta_1$ | 0.9 |
| $\beta_2$ | 0.999 |
| Weight decay | $10^{-5}$ |
| Batch size | 64 |
| Training epochs | 200 |
| Warm up percentage | 10% |
| Learning rate decay | Cosine |

## C.2 Baselines

We train baseline algorithms following their officially released configurations and tune the learning rate, learning rate scheduler, working resolution, etc., to optimize their performance on each dataset. We list the modifications we applied to the baselines for each dataset in Table 13.

Table 13: Implementation details of baseline algorithms. Modifications based on the officially released implementations are listed according to different datasets. "-" means no modification is applied. "reverse enc-dec" means adopting the reversed encoder-decoder architecture proposed in [48]. Other terms listed are the hyperparameters in their officially released implementations.

| Model | $N$-body MNIST | SEVIR |
|---|---|---|
| UNet [55] | - | - |
| ConvLSTM [47] | reverse enc-dec [48]
conv_kernels = [(7,7),(5,5),(3,3)]
deconv_kernels = [(6,6),(4,4),(4,4)]
channels=[96, 128, 256] | reverse enc-dec [48]
conv_kernels = [(7,7),(5,5),(3,3)]
deconv_kernels = [(6,6),(4,4),(4,4)]
channels=[96, 128, 256] |
| PredRNN [61] | - | - |
| PhyDNet [11] | - | convcell_hidden = [256, 256, 256, 64] |
| E3D-LSTM [60] | - | - |
| Rainformer [1] | downscaling_factors=[2, 2, 2, 2]
hidden_dim=32
heads=[4, 4, 8, 16]
head_dim=8 | downscaling_factors=[4, 2, 2, 2]
-
-
- |
| Earthformer [8] | - | - |
| DGMR [41] | - | context_steps = 7
forecast_steps = 6 |
| VideoGPT [65] | vqvae_n_codes = 512
vqvae_downsample = [1, 4, 4] | vqvae_n_codes = 512
vqvae_downsample = [1, 8, 8] |
| LDM [42] | vae: $64 \times 64 \times 1 \rightarrow 16 \times 16 \times 3$
conv_dim = 3
model_channels = 256 | vae: $128 \times 128 \times 1 \rightarrow 16 \times 16 \times 4$
conv_dim = 3
model_channels = 256 |

## D   Derivation of the Approximation to Knowledge Alignment Guidance

We derive the approximation to the knowledge alignment guided denoising transition (5) following [4]. We rewrite (5) to (8) using a normalization constant $Z$ that normalizes $Z \int e^{-\lambda_{\mathcal{F}} \|U_\phi(z_t, t, y) - \mathcal{F}_0(y)\|} dz_t = 1$:

$$p_{\theta,\phi}(z_t | z_{t+1}, y, \mathcal{F}_0) = p_\theta(z_t | z_{t+1}, z_{\text{cond}}) \cdot Z e^{-\lambda_{\mathcal{F}} \|U_\phi(z_t, t, y) - \mathcal{F}_0(y)\|}. \tag{8}$$

In what follows, we abbreviate $\mu_\theta(z_{t+1}, t, z_{\text{cond}})$ as $\mu_\theta$, and $\Sigma_\theta(z_{t+1}, t, z_{\text{cond}})$ as $\Sigma_\theta$ for brevity. We use $C_i, i = \{1, \dots, 7\}$ to denote constants.

$$p_\theta(z_t | z_{t+1}, z_{\text{cond}}) = \mathcal{N}(\mu_\theta, \Sigma_\theta),$$

$$\log p_\theta(z_t | z_{t+1}, z_{\text{cond}}) = -\frac{1}{2}(z_t - \mu_\theta)^T \Sigma_\theta^{-1}(z_t - \mu_\theta) + C_1, \tag{9}$$

$$\log Z e^{-\lambda_{\mathcal{F}} \|U_\phi(z_t, t, y) - \mathcal{F}_0(y)\|} = -\lambda_{\mathcal{F}} \|U_\phi(z_t, t, y) - \mathcal{F}_0(y)\| + C_2,$$

By assuming that $\log Z e^{-\lambda_{\mathcal{F}} \|U_\phi(z_t, t, y) - \mathcal{F}_0(y)\|}$ has low curvature compared to $\Sigma_\theta^{-1}$, which is reasonable in the limit of infinite diffusion steps ($\|\Sigma_\theta\| \to 0$), we can approximate it by a Taylor expansion at $z_t = \mu_\theta$

$$\begin{aligned}
\log Z e^{-\lambda_{\mathcal{F}} \|U_\phi(z_t, t, y) - \mathcal{F}_0(y)\|} &\approx -\lambda_{\mathcal{F}} \|U_\phi(z_t, t, y) - \mathcal{F}_0(y)\| \big|_{z_t = \mu_\theta} \\
&\quad - (z_t - \mu_\theta) \lambda_{\mathcal{F}} \nabla_{z_t} \|U_\phi(z_t, t, y) - \mathcal{F}_0(y)\| \big|_{z_t = \mu_\theta} \\
&= (z_t - \mu_\theta) g + C_3,
\end{aligned} \tag{10}$$

where $g = -\lambda_{\mathcal{F}} \nabla_{z_t} \|U_\phi(z_t, t, y) - \mathcal{F}_0(y)\| \big|_{z_t = \mu_\theta}$. By taking the log of (8) and applying the results from (9) and (10), we get

$$\begin{aligned}
\log p_{\theta,\phi}(z_t | z_{t+1}, y, \mathcal{F}_0) &= \log p_\theta(z_t | z_{t+1}, z_{\text{cond}}) + \log Z e^{-\lambda_{\mathcal{F}} \|U_\phi(z_t, t, y) - \mathcal{F}_0(y)\|} \\
&\approx -\frac{1}{2}(z_t - \mu_\theta)^T \Sigma_\theta^{-1}(z_t - \mu_\theta) + (z_t - \mu_\theta) g + C_4 \\
&= -\frac{1}{2}(z_t - \mu_\theta - \Sigma_\theta g)^T \Sigma_\theta^{-1}(z_t - \mu_\theta - \Sigma_\theta g) + \frac{1}{2} g^T \Sigma_\theta g + C_5 \\
&= -\frac{1}{2}(z_t - \mu_\theta - \Sigma_\theta g)^T \Sigma_\theta^{-1}(z_t - \mu_\theta - \Sigma_\theta g) + C_6 \\
&= \log p(z) + C_7, \quad z \sim \mathcal{N}(\mu_\theta + \Sigma_\theta g, \Sigma_\theta).
\end{aligned} \tag{11}$$

Therefore, the transition distribution under the guidance of knowledge alignment shown in (5) can be approximated by a Gaussian similar to the transition without knowledge guidance, but with its mean shifted by $\Sigma_\theta$.

# E More Quantitative Results on SEVIR

## E.1 Quantitative Analysis of `BIAS` on SEVIR

Similar to `Critical Success Index` (CSI) introduced in Sec. 3.1, $\text{BIAS} = \frac{\#\text{Hits}+\#\text{F.Alarms}}{\#\text{Hits}+\#\text{Misses}}$ is calculated by counting the #Hits (truth=1, pred=1), #Misses (truth=1, pred=0) and #F.Alarms (truth=0, pred=1) of the predictions binarized at thresholds $[16, 74, 133, 160, 181, 219]$. This measurement assesses the model's inclination towards either F.Alarms or Misses.

The results from Table 14 demonstrate that deterministic spatiotemporal forecasting models, such as UNet [55], ConvLSTM [47], PredRNN [61], PhyDNet [11], E3D-LSTM [60], and Earthformer [8], tend to produce predictions with lower intensity. These models prioritize avoiding high-intensity predictions that have a higher chance of being incorrect due to their limited ability to handle such uncertainty effectively. On the other hand, probabilistic spatiotemporal forecasting baselines, including DGMR [41], VideoGPT [65] and LDM [42], demonstrate a more daring approach by predicting possible high-intensity signals, even if it results in lower CSI scores, as depicted in Table 1. Among these baselines, PreDiff achieves the best performance in BIAS. It consistently achieves BIAS scores closest to 1, irrespective of the chosen threshold. These results demonstrate that PreDiff has effectively learned to unbiasedly capture the distribution of intensity.

Table 14: Quantitative Analysis of `BIAS` on SEVIR. The `BIAS` is calculated at different precipitation thresholds and denoted as `BIAS`-$thresh$. `BIAS-m` reports the mean of `BIAS`-$[16, 74, 133, 160, 181, 219]$. A `BIAS` score closer to 1 indicates that the model is less biased to either F.Alarms or Misses. The best `BIAS` score is in boldface while the second best is underscored.

| Model | Metrics | | | | | | |
|---|---|---|---|---|---|---|---|
| | BIAS-m | BIAS-219 | BIAS-181 | BIAS-160 | BIAS-133 | BIAS-74 | BIAS-16 |
| Persistence | 1.0177 | 1.0391 | 1.0323 | 1.0258 | 1.0099 | 1.0016 | 0.9983 |
| UNet [55] | 0.6658 | 0.2503 | 0.4013 | 0.5428 | 0.7665 | 0.9551 | 1.0781 |
| ConvLSTM [47] | 0.8341 | 0.5344 | 0.6811 | 0.7626 | 0.9643 | **0.9957** | 1.0663 |
| PredRNN [61] | 0.6605 | 0.2565 | 0.4377 | 0.4909 | 0.6806 | 0.9419 | 1.1554 |
| PhyDNet [11] | 0.6798 | 0.3970 | 0.6593 | 0.7312 | 1.0543 | 1.0553 | 1.2238 |
| E3D-LSTM [60] | 0.6925 | 0.2696 | 0.4861 | 0.5686 | 0.8352 | 0.9887 | 1.0070 |
| Earthformer [8] | 0.7043 | 0.2423 | 0.4605 | 0.5734 | 0.8623 | 0.9733 | 1.1140 |
| DGMR [41] | 0.7302 | 0.3704 | 0.5254 | 0.6495 | 0.8312 | 0.9594 | 1.0456 |
| VideoGPT [65] | 0.8594 | 0.6106 | 0.7738 | 0.8629 | 0.9606 | 0.9681 | 0.9805 |
| LDM [42] | 1.2951 | 1.4534 | 1.3525 | 1.3827 | 1.3154 | 1.1817 | 1.0847 |
| PreDiff | **0.9769** | **0.9647** | **0.9268** | **0.9617** | **0.9978** | 1.0047 | **1.0058** |

## E.2  `CSI` at Varying Thresholds on SEVIR

We include representative deterministic methods ConvLSTM and Earthformer, and all studied probabilistic methods to compare `CSI`, `CSI`, `CSI-pool4` and `CSI-pool16` at varying thresholds. It is important to note that `CSI` tends to favor conservative predictions, especially in situations with high levels of uncertainty. To ensure a fair comparison, we calculated the `CSI` scores by averaging the samples for each model, while scores in other metrics are averaged over the scores of each sample. The results presented in Table 15, 16, 17 demonstrate that our PreDiff achieves competitive `CSI` scores and outperforms baselines in `CSI` scores at pooling scale $4 \times 4$ and $16 \times 16$, particularly at higher thresholds.

Table 15: `CSI` at thresholds $[16, 74, 133, 160, 181, 219]$ on SEVIR.

| Model | Metrics | | | | | | |
|---|---|---|---|---|---|---|---|
| | CSI-m ↑ | CSI-219 ↑ | CSI-181 ↑ | CSI-160 ↑ | CSI-133 ↑ | CSI-74 ↑ | CSI-16 ↑ |
| ConvLSTM [47] | 0.4185 | 0.1220 | 0.2381 | 0.2905 | 0.4135 | 0.6846 | 0.7510 |
| Earthformer [8] | **0.4419** | **0.1791** | **0.2848** | **0.3232** | **0.4271** | **0.6860** | **0.7513** |
| DGMR [41] | 0.2675 | 0.0151 | 0.0537 | 0.0970 | 0.2184 | 0.5500 | 0.6710 |
| VideoGPT [65] | 0.3653 | 0.1029 | 0.1997 | 0.2352 | 0.3432 | 0.6062 | 0.7045 |
| LDM [42] | 0.3580 | 0.1019 | 0.1894 | 0.2340 | 0.3537 | 0.5848 | 0.6841 |
| PreDiff | 0.4100 | 0.1154 | 0.2357 | 0.2848 | 0.4119 | 0.6740 | 0.7386 |

Table 16: `CSI-pool4` at thresholds $[16, 74, 133, 160, 181, 219]$ on SEVIR.

| Model | Metrics | | | | | | |
|---|---|---|---|---|---|---|---|
| | CSI-pool4-m ↑ | CSI-pool4-219 ↑ | CSI-pool4-181 ↑ | CSI-pool4-160 ↑ | CSI-pool4-133 ↑ | CSI-pool4-74 ↑ | CSI-pool4-16 ↑ |
| ConvLSTM [47] | 0.4452 | 0.1850 | 0.2864 | 0.3245 | 0.4502 | 0.6694 | 0.7556 |
| Earthformer [8] | 0.4567 | 0.1484 | 0.2772 | 0.3341 | **0.4911** | **0.7006** | **0.7892** |
| DGMR [41] | 0.3431 | 0.0414 | 0.1194 | 0.1950 | 0.3452 | 0.6302 | 0.7273 |
| VideoGPT [65] | 0.4349 | 0.1691 | 0.2825 | 0.3268 | 0.4482 | 0.6529 | 0.7300 |
| LDM [42] | 0.4022 | 0.1439 | 0.2420 | 0.2964 | 0.4171 | 0.6139 | 0.6998 |
| PreDiff | **0.4624** | **0.2065** | **0.3130** | **0.3613** | 0.4807 | 0.6691 | 0.7438 |

Table 17: `CSI-pool16` at thresholds $[16, 74, 133, 160, 181, 219]$ on SEVIR.

| Model | Metrics | | | | | | |
|---|---|---|---|---|---|---|---|
| | CSI-pool16-m ↑ | CSI-pool16-219 ↑ | CSI-pool16-181 ↑ | CSI-pool16-160 ↑ | CSI-pool16-133 ↑ | CSI-pool16-74 ↑ | CSI-pool16-16 ↑ |
| ConvLSTM [47] | 0.5135 | 0.2651 | 0.3679 | 0.4153 | 0.5408 | 0.7039 | 0.7883 |
| Earthformer [8] | 0.5005 | 0.1798 | 0.3207 | 0.3918 | 0.5448 | 0.7304 | **0.8353** |
| DGMR [41] | 0.4832 | 0.1218 | 0.2804 | 0.3924 | 0.5364 | 0.7465 | 0.8216 |
| VideoGPT [65] | 0.5798 | 0.3101 | 0.4543 | 0.5211 | 0.6285 | 0.7583 | 0.8065 |
| LDM [42] | 0.5522 | 0.2896 | 0.4247 | 0.4987 | 0.5895 | 0.7229 | 0.7876 |
| PreDiff | **0.6244** | **0.3865** | **0.5127** | **0.5757** | **0.6638** | **0.7789** | 0.8289 |

# F   More Qualitative Results on $N$-body MNIST

Fig. 6 to Fig. 13 show several sets of example predictions on the $N$-body MNIST test set. In each figure, visualizations from top to bottom are context sequence $y$, target sequence $x$, predictions by ConvLSTM [47], Earthformer [8], VideoGPT [65], LDM [42], PreDiff, PreDiff-KA. E.MSE denotes the average error between the total energy (the sum of kinetic energy and potential energy) of the predictions $E(\widehat{x}^j)$ and the total energy of the last step context $E(y^{L_{\mathrm{in}}})$.

Figure 6: A set of example predictions on the $N$-body MNIST test set. The red dashed line is to help the reader to judge the position of the digit "1" in the last frame.

Figure 7: A set of example predictions on the $N$-body MNIST test set. The red dashed line is to help the reader to judge the position of the digit "0" in the last frame.

Figure 8: A set of example predictions on the $N$-body MNIST test set. The red dashed line is to help the reader to judge the position of the digit "0" in the last frame.

Figure 9: A set of example predictions on the $N$-body MNIST test set. The red dashed line is to help the reader to judge the position of the digit "8" in the last frame.

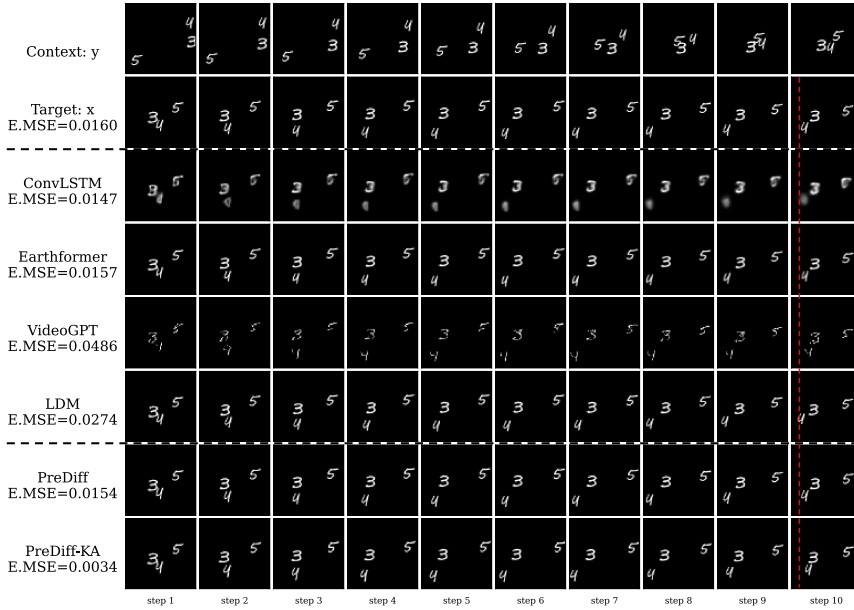

Figure 10: A set of example predictions on the $N$-body MNIST test set. The red dashed line is to help the reader to judge the position of the digit "4" in the last frame.

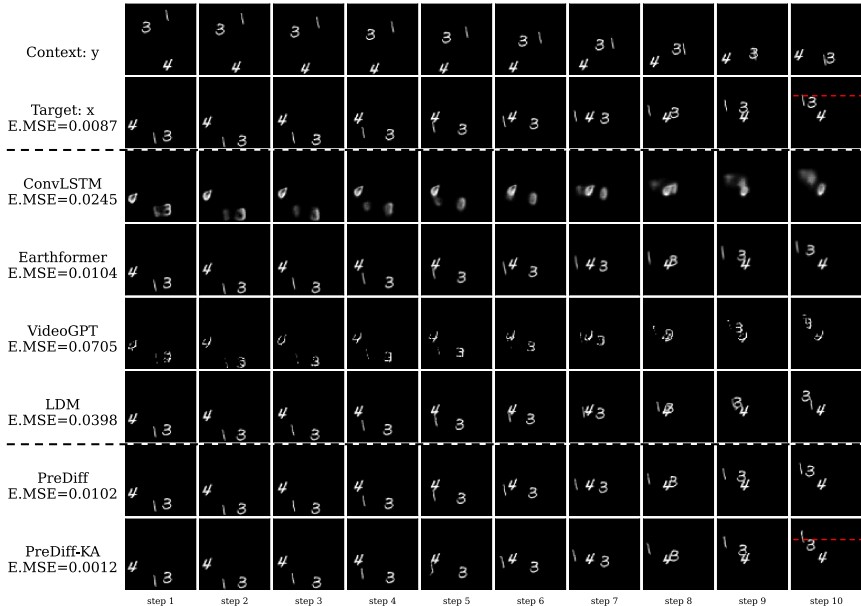

Figure 11: A set of example predictions on the $N$-body MNIST test set. The red dashed line is to help the reader to judge the position of the digit "1" in the last frame.

Figure 12: A set of example predictions on the $N$-body MNIST test set. The red dashed line is to help the reader to judge the position of the digit "7" in the last frame.

Figure 13: A set of example predictions on the $N$-body MNIST test set. The red dashed line is to help the reader to judge the position of the digit "7" in the last frame.

# G  More Qualitative Results on SEVIR

Fig. 14 to Fig. 19 show several sets of example predictions on the SEVIR test set. In subfigure (a) of each figure, visualizations from top to bottom are context sequence $y$, target sequence $x$, predictions by ConvLSTM [47], Earthformer [8], VideoGPT [65], LDM [42], PreDiff, PreDiff-KA. In subfigure (b) of each figure, visualizations from top to bottom are context sequence $y$, target sequence $x$, predictions by PreDiff-KA with anticipated average future intensity $\mu_\tau + n\sigma_\tau$, $n = 4, 2, 0 - 2, -4$.

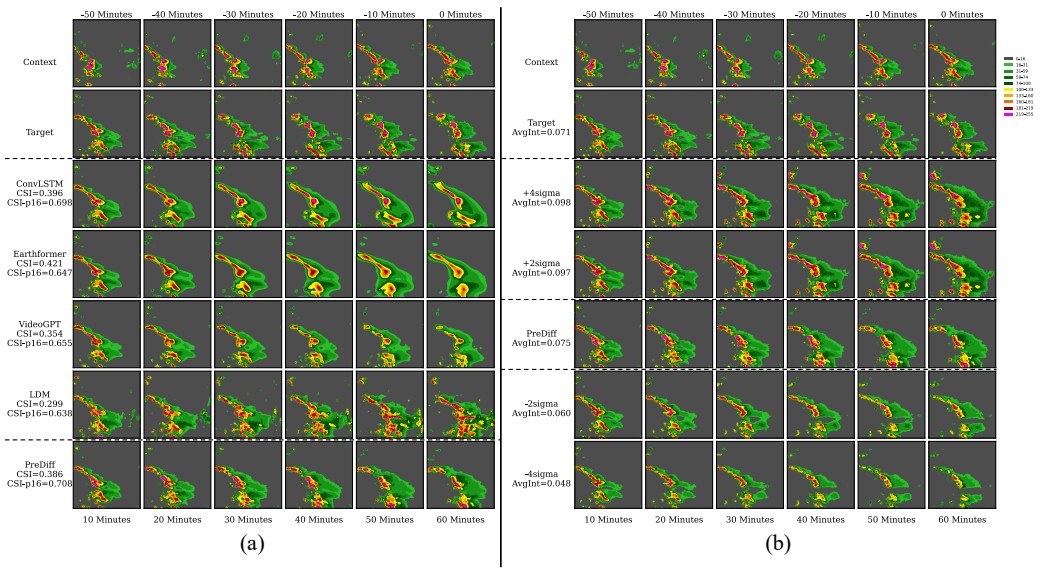

Figure 14: A set of example predictions on the SEVIR test set. (a) Comparison of PreDiff with baselines. (b) Predictions by PreDiff-KA under the guidance of anticipated average intensity.

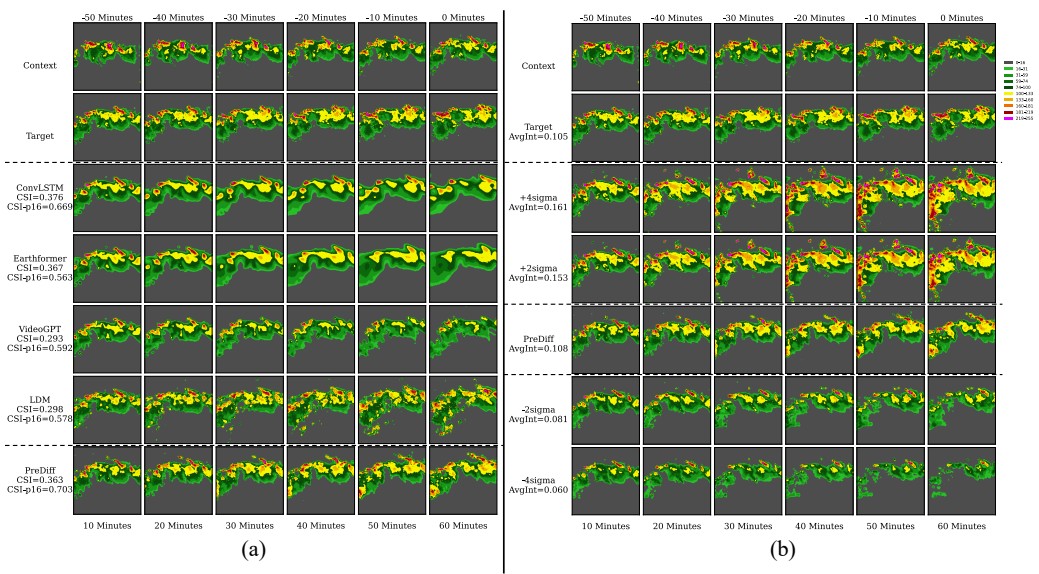

Figure 15: A set of example predictions on the SEVIR test set. (a) Comparison of PreDiff with baselines. (b) Predictions by PreDiff-KA under the guidance of anticipated average intensity.

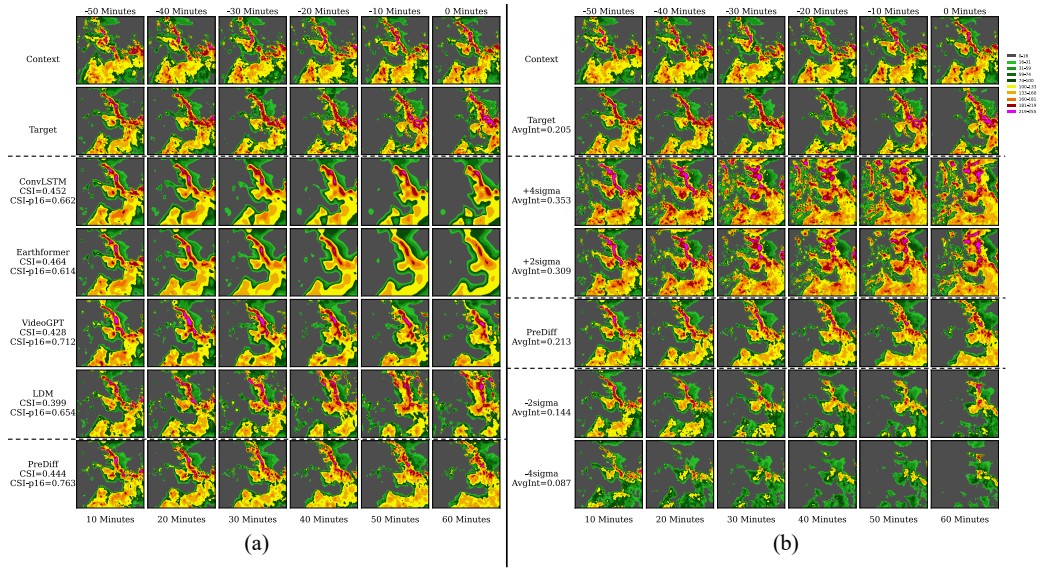

Figure 16: A set of example predictions on the SEVIR test set. (a) Comparison of PreDiff with baselines. (b) Predictions by PreDiff-KA under the guidance of anticipated average intensity.

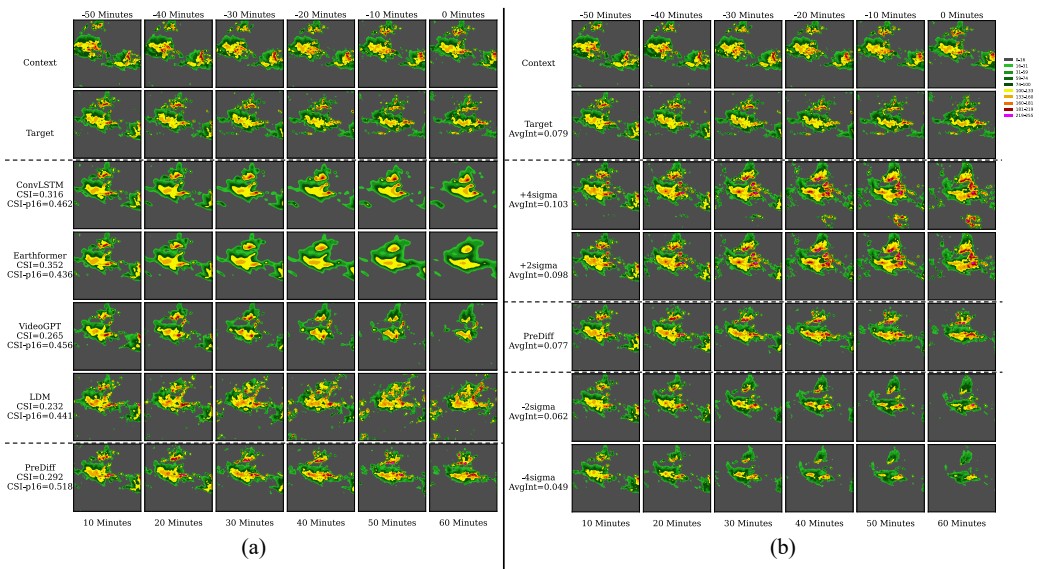

Figure 17: A set of example predictions on the SEVIR test set. (a) Comparison of PreDiff with baselines. (b) Predictions by PreDiff-KA under the guidance of anticipated average intensity.

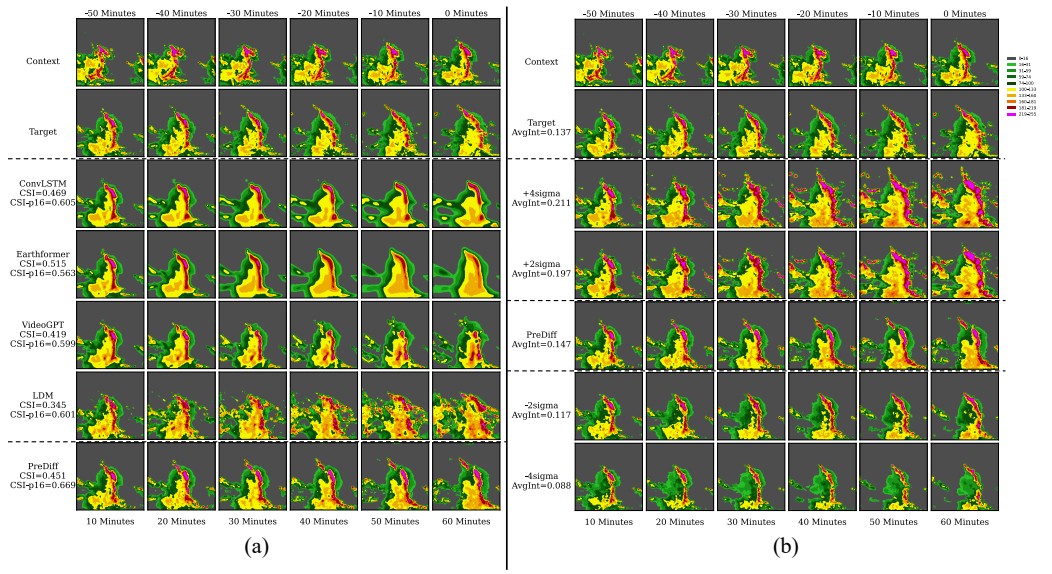

Figure 18: A set of example predictions on the SEVIR test set. (a) Comparison of PreDiff with baselines. (b) Predictions by PreDiff-KA under the guidance of anticipated average intensity.

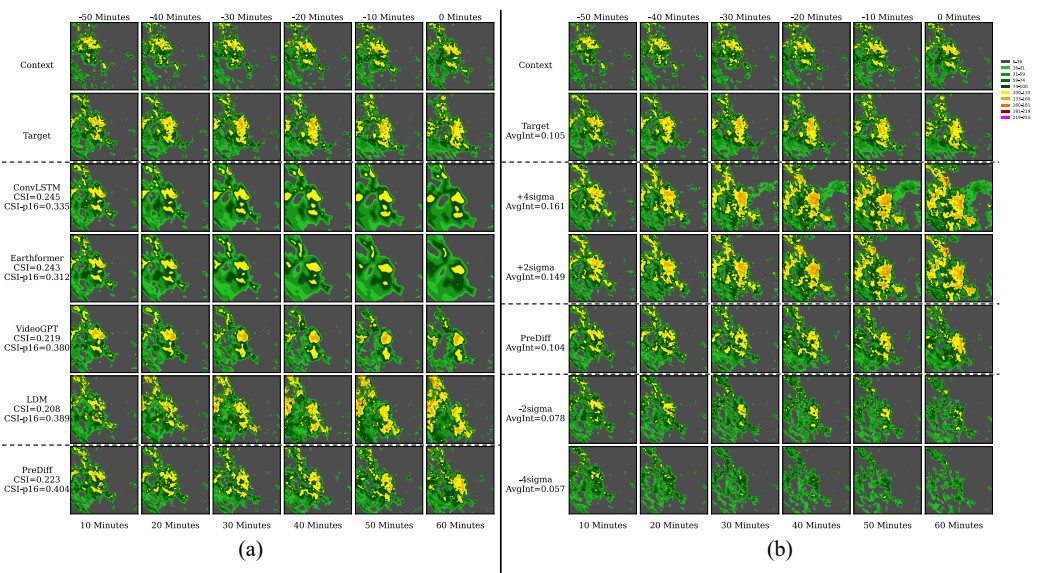

Figure 19: A set of example predictions on the SEVIR test set. (a) Comparison of PreDiff with baselines. (b) Predictions by PreDiff-KA under the guidance of anticipated average intensity.

