# OpenReview forum: "PreDiff: Precipitation Nowcasting with Latent Diffusion Models"
_NeurIPS.cc/2023/Workshop/AI4Science — NeurIPS2023-AI4Science Poster_

### Official Review · Reviewer_pB7v · 2023-10-19
**Potentially impactful work on probabilistic precipitation forecasting**

**Rating:** 8
**Confidence:** 3

**Review:**

**Strengths**

**S1:** The paper takes a probabilistic perspective on precipitation forecasting. It is true that the chaotic nature of Earth system dynamics leads to a range of strongly diverging plausible futures. Consequently, a diffusion model as proposed by the authors is a sensible match to this problem. Overall, the presented modeling framework is technically sound and well-designed for the problem at hand.

**S2:** The knowledge control framework is an elegant and generally applicable framework for incorporating domain knowledge into LDM models for spatiotemporal forecasting. Due to its generality, knowledge control could have a larger impact on the broader community.

**S3:**  The method and results are clearly presented and should be easy to follow for members of the AI4Sciece community.

---

**Comments**

**C1:** I found the discussion of the related work on weather forecasting models insufficient. The authors do point to references on the state-of-the-art in their introduction, but I missed an explanation of the key differences between PreDiff and other approaches, among which probabilistic methods in particular.

---

**Minor comments, questions and suggestions**

**C2:** Could knowledge control also be extended so that it is applicable to model architectures beyond diffusion models? Can the authors elaborate on possible broader impacts of the knowledge control framework beyond the precipitation forecasting domain explored in this paper?

**C3:** The authors mention the importance of integrating domain knowledge into spatiotemporal forecasting models. Although using knowledge control is an elegant and general way to do this, it does not offer hard guarantees with respect to the quantities that need to be conserved. To this end, I wonder if the authors have also considered the incorporation of geometrical symmetries into their forecasting model, e.g. see for example [1, 2, 3], as this has also been shown to improve forecast performance and generalization. I understand that this is out of scope for this paper, but the authors’ perspective on this and how they envision the incorporation of known symmetries relative to the knowledge control framework would be a great addition for a future work section.

**C4:** The footnote on the first page mentions the wrong year.

---

**Conclusion**

Overall, I found this an interesting and well-executed paper which applies LDM to precipitation nowcasting. Besides the application, the paper also comes with an interesting technical contribution in the form of the knowledge control framework, which could have a broader impact beyond the weather forecasting community. The results demonstrate the strong performance of the model.
Apart from C1, I only had minor comments on this work, and I believe the paper would be an excellent addition to the AI4Science workshop. I encourage the authors to address the comments as much as possible in the revised version of the paper.

---

**References**

[1] Bonev, B., Kurth, T., Hundt, C., Pathak, J., Baust, M., Kashinath, K., & Anandkumar, A. (2023). Spherical Fourier Neural Operators: Learning Stable Dynamics on the Sphere. arXiv preprint arXiv:2306.03838. ICML 2023.

[2] Wang, R., Walters, R., & Yu, R. (2021). Incorporating Symmetry into Deep Dynamics Models for Improved Generalization. ICLR 2021.

[3] Minartz, K., Poels, Y., Koop, S., & Menkovski, V. (2023). Equivariant Neural Simulators for Stochastic Spatiotemporal Dynamics. arXiv preprint arXiv:2305.14286.

---

### Meta-Review · Area_Chair_kLFf · 2023-10-27

**Recommendation:** Accept (Poster)
**Confidence:** 4

**Metareview:**

The authors describe a latent diffusion model for precipitation forecasting building on methods which bring a probabilistic perspective to the problem. They report improved performance over current methods on various other metrics which align closer to perceptual quality such as FVD and pooled CSI, but do not exceed SOTA on the CSI metric. They introduce a form of classifier-based guidance called knowledge control which aligns diffusion samples with domain knowledge and may be useful outside the weather domain.